# Sparse Topology-Aware Pairwise Scoring for Large-Scale Multi-Agent Reinforcement Learning

**Zhibo Deng** [1 2]  **Feng Liang** [1 2]  **Yong Zhang** [1 2 3]  **Xiaoxi Zhang** [4 5]  **Xiping Hu** [1 2 3]

## Abstract

In multi-agent reinforcement learning (MARL), communication enables agents to mitigate partial observability and stochasticity through information sharing, but large-scale systems inherently lead to a rapidly growing number of pairwise interactions. Previous studies often struggle to simultaneously achieve scalability and task adaptivity in large-scale multi-agent communication. To address this challenge, we propose a scalable communication scheme for large-scale MARL, termed *Sparse tOpology-aware Pairwise Scoring* (SOPS). We argue that scalable MARL communication requires decoupling scalability from task-adaptive link allocation. To ensure scalability, we constrain communication to an exponential-graph backbone with a small diameter, which preserves rapid potential information mixing while keeping per-agent candidates logarithmic. On top of this constraint, we learn a task-conditioned probabilistic subgraph distribution via a pairwise scoring network over agent states and edge-type embeddings to allocate sparse links for maximizing return, optimized end-to-end through differentiable Gumbel-Sigmoid reparameterization. Evaluation results show that SOPS significantly outperforms existing state-of-the-art methods across cooperative benchmarks of diverse scales and exhibits robust zero-shot transfer capabilities.

[1]Artificial Intelligence Research Institute, Shenzhen MSU-BIT University [2]Guangdong–Hong Kong–Macau Joint Laboratory for Emotional Intelligence and Pervasive Computing, Shenzhen MSU-BIT University [3]School of Medical Technology, Beijing Institute of Technology [4]School of Computer Science and Engineering, Sun Yat-sen University [5]Shenzhen Loop Area Institute. Correspondence to: Feng Liang <fliang@smbu.edu.cn>, Xiping Hu <huxp@smbu.edu.cn>.

*Proceedings of the 43rd International Conference on Machine Learning*, Seoul, South Korea. PMLR 306, 2026. Copyright 2026 by the author(s).

## 1. Introduction

Cooperative multi-agent reinforcement learning (MARL) has achieved significant advancements in a variety of complex decision-making applications, including complex virtual games (Barambones et al., 2022), simulated industrial control (Wang et al., 2022b; Leroy et al., 2023), autonomous driving (Dinneweth et al., 2022) and robotic planning (Zhang et al., 2021). These scenarios are often constrained by non-stationarity and partial observability (Omidshafiei et al., 2017). A widely adopted solution is the centralized training with decentralized execution (CTDE) paradigm, which leverages global information to train value functions or critics while constraining each agent to local observations at test time (Kraemer & Banerjee, 2016; Lowe et al., 2017). Building upon the CTDE framework, explicit inter-agent communication facilitates improved coordination by allowing agents to exchange mission-critical information that extends beyond their local observations. Recent studies have demonstrated that communication significantly improves decision-making efficiency and robustness across diverse cooperative scenarios, including group-aware coordination (Duan et al., 2024), targeted and trusted message exchange (Sun et al., 2024b), personalized communication (Meng & Tan, 2024), and intention-aware protocols inspired by the theory of mind (Wang et al., 2021). These advances show the crucial role of communication schemes in scaling MARL to more complex and dynamic environments.

Despite the steady progress, existing CTDE methods become increasingly impractical as the number of agents grows, falling into the curse of dimensionality (Oroojlooy & Hajinezhad, 2023). This issue has been widely recognized in recent work on large-population systems and many-agent MARL (Cui et al., 2022; He et al., 2022). They typically rely on global state or observation information, training with which usually leads to sharply declining performance and poor sample efficiency at large scales (Wang et al., 2023). We identify two key challenges in designing such scalable communication mechanisms for the CTDE paradigm: achieving communication efficiency at large scales (scalability) and ensuring the task adaptivity of methods to various scales (adaptability). First, even in decentralized settings,

dense or all-to-all communication patterns generate excessive message traffic and activation storage, quickly exceeding available bandwidth and GPU memory (Das et al., 2019; Iqbal & Sha, 2019a), regardless of centralized or decentralized settings. Second, learned communication structures tend to be fragile, exhibiting instability during training or poor generalization when applied to different agent populations or dynamic environments (Du et al., 2021; Hu et al., 2021).

Recently, some studies have attempted to scale MARL systems with reduced communication, partially addressing these challenges. For example, Chiun et al. (2025) enhance large-scale multi-robot exploration by pruning Graph Attention Networks and actions based on frontier-based information gain to avoid all-to-all communication. GTDE (Li et al., 2025a) learns adaptive groups for sharing information during training, stabilizing training through intra-group cross-agent information and mitigating non-stationarity. However, both methods start off with a full communication topology, and the resulting communication bandwidth is not properly bounded. Moreover, due to the limitations imposed by its network architecture design, GTDE's grouping strategy, learned from an agent population, cannot be seamlessly applied to another population scale. Li et al. (2025b) introduce a rule-based exponential communication topology, whose small-diameter communication is proven with bounded bandwidth for various scales. However, this fixed topology has difficulty in adapting communication links to task dynamics, thereby reducing knowledge exchange efficiency.

In this work, we propose *Sparse tOpology Pairwise Scoring* (SOPS), a scalable, sparse communication mechanism that separates global reachability from task-adaptive selection in the CTDE paradigm. Concretely, we instantiate an exponential graph as a small-diameter backbone, which guarantees fast multi-hop reachability under tight bandwidth and keeps the communication cost near-linear in the number of agents. On top of this backbone, a lightweight pairwise-scoring module selects a few peers per agent at each timestep via Gumbel-based sampling (Jang et al., 2017) with a linearly annealed temperature, yielding a task-adaptive, time-varying communication graph. Messages are aggregated with cross-attention blocks to accumulate multi-hop information, and simple auxiliary objectives are used to ground the messages during training. SOPS is pluggable with common value-based learners (e.g., QMIX (Samvelyan et al., 2019) / QPLEX (Wang et al., 2020)) and executes fully decentralized without a proxy. Through extensive experiments in various cooperative MARL scenarios of various scales, SOPS outperforms other state-of-the-art (SOTA) methods by consistently achieving higher returns and faster convergence. It also exhibits robust zero-shot transfer ability to larger agent populations. Our contributions can be summa-

rized as follows:

- SOPS is efficient and scalable. Its lightweight pairwise scoring enables budget-aware dynamic links in the exponential topology-based communication for large-scale MARL.
- SOPS is adaptable to various scales. Gumbel sampling and linear annealing allow end-to-end training and deployment across varying population sizes without topology redesign or population-specific retraining.
- SOPS demonstrates SOTA performance in various large-scale cooperative scenarios. It achieves higher returns and faster convergence, and has robust zero-shot transfer ability.

## 2. Related Work

### 2.1. Multi-agent Cooperation Paradigms

The study of cooperation in MARL has evolved rapidly, with a variety of paradigms proposed to enable effective coordination in partially observable environments (Foerster et al., 2016). Central to this progress is the CTDE paradigm, which leverages global state information during training while enabling agents to act independently based on local observations at test time (Oliehoek et al., 2016; Lowe et al., 2017). Extensions of CTDE, such as value decomposition methods (Sunehag et al., 2017; Samvelyan et al., 2019; Wang et al., 2020), aim to decompose the global team reward into local value functions, improving scalability and interpretability. Beyond reward decomposition, advances in credit assignment and counterfactual reasoning (Foerster et al., 2018) align individual behavior with team objectives under partial observability. Other lines of work investigate hierarchical cooperation frameworks (Yang et al., 2019; Iqbal et al., 2022), where agents operate under high-level coordination strategies, enabling task abstraction and modular training. Despite these developments, challenges remain in generalization and scalability to large agent populations. Kontogiannis et al. (2025) propose SMPE2, a state modeling framework that enables agents to infer agent-specific belief representations under partial observability and leverages them for adversarial exploration to enhance policy learning.

### 2.2. Communication Learning and Topology Design

Effective communication among agents plays a crucial role in improving coordination, especially in partially observable or dynamic environments. Early work introduced differentiable communication channels, enabling agents to learn end-to-end message exchange during training (Sukhbaatar et al., 2016; Das et al., 2019). Later methods leverage attention mechanisms to allow agents to selectively aggregate information from relevant peers, improving efficiency and interpretability (Iqbal & Sha, 2019b). Beyond continuous

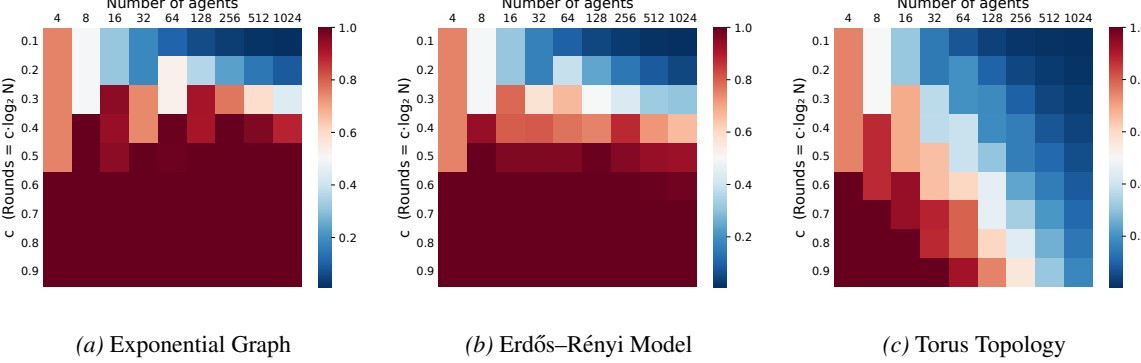

*(a)* Exponential Graph      *(b)* Erdős–Rényi Model      *(c)* Torus Topology

*Figure 1.* Comparison of log-step broadcast coverage under a unified per-round bandwidth. Columns give the number of agents, rows give a dimensionless time scale $c$, which sets the number of synchronous rounds $S = \lceil c \log_2 N \rceil$. Each cell shows the fraction of agents informed after $S$ rounds when every informed node may contact at most $K(N) = \lceil \log_2 N \rceil$ neighbors per round (if a node's degree is $< K$, it contacts all of its neighbors).

message passing, discrete protocols and emergent language have been explored to co-evolve communication and task policies (Eccles et al., 2019; Chafii et al., 2023). Recent studies such as M2I2 (Sun et al., 2024a) formulate communication as an information-integration problem and introduce self-supervised objectives, including masked state modeling for reconstructing missing global information and joint-action inference, to make message usage under partial observability and bandwidth constraints.

# 3. Preliminaries

## 3.1. Cooperative MARL Problems Formulation

We model cooperative multi-agent reinforcement learning (MARL) tasks as a decentralized partially observable Markov decision process (Dec-POMDP) (Oliehoek et al., 2016). It is defined by the 8-tuple $\mathcal{M} = \langle \mathcal{I}, \mathcal{S}, \{\mathcal{A}_i\}_{i \in \mathcal{I}}, \mathcal{P}, \mathcal{R}, \{\Omega_i\}_{i \in \mathcal{I}}, \mathcal{O}, \gamma \rangle$, where $\mathcal{I} = \{1, \ldots, n\}$ is the set of agents, $\mathcal{S}$ is the global state space with $s^t \in \mathcal{S}$, and $\mathcal{A}_i$ is the action space of agent $i$. The joint action is $a^t = (a_1^t, \ldots, a_n^t) \in \prod_{i \in \mathcal{I}} \mathcal{A}_i$, the transition kernel is $\mathcal{P}(s^{t+1} \mid s^t, a^t)$, and the shared team reward is $r^t = \mathcal{R}(s^t, a^t)$. Each agent $i$ receives a local observation $o_i^t \in \Omega_i$, and the joint observation $o^t = (o_1^t, \ldots, o_n^t) \in \prod_{i \in \mathcal{I}} \Omega_i$ is generated by the observation function $\mathcal{O}(o^{t+1} \mid s^{t+1}, a^t)$. Due to partial observability, agent $i$ conditions on its local action-observation history $\eta_i^t = (o_i^{1:t}, a_i^{1:t-1})$ and follows a decentralized policy $\pi_{\theta_i}(a_i^t \mid \eta_i^t)$. The objective is to learn decentralized policies $\pi = \{\pi_{\theta_i}\}_{i \in \mathcal{I}}$ that maximize the expected discounted return $J(\pi) = \mathbb{E}_{s^0 \sim \rho, \mathcal{P}, \mathcal{O}, \pi} \left[ \sum_{t=0}^{T-1} \gamma^t r^t \right]$, where $\rho$ is the initial state distribution. This study focuses on the dynamic generation of $\mathcal{E}^t$ for each timestep $t$ to address efficient communication in large-scale cooperative MARL.

## 3.2. Communication model.

When explicit communication is available, we represent the inter-agent connectivity at time $t$ by a (possibly time-varying) directed graph $\mathcal{G}^t = (\mathcal{I}, \mathcal{E}^t)$, where $(j \to i) \in \mathcal{E}^t$ indicates that agent $i$ can receive a message from agent $j$. We denote the in-neighborhood of agent $i$ as $\mathcal{N}_i^t = \{j \mid (j \to i) \in \mathcal{E}^t\}$. At each step, agent $i$ aggregates messages from its neighbors (from the previous step) into a context vector $c_i^t = \text{Agg}_i \{m_j^{t-1} : j \in \mathcal{N}_i^t\}$, and updates its communication message as $m_i^t = \text{Msg}(h_i^t, c_i^t)$, where $h_i^t$ is the agent's internal representation (e.g., a recurrent hidden state derived from $\eta_i^t$). The action policy can then be conditioned on the communication signal, e.g., $\pi_{\theta_i}(a_i^t \mid \eta_i^t, m_i^t)$. We study how to generate $\mathcal{E}^t$ within a scalable candidate domain while maintaining task adaptivity.

# 4. Methodology

Motivated by the scalability and adaptability trade-off in large-scale MARL, we formulate SOPS as a hierarchical communication design under the CTDE framework that decouples feasible global reachability from task-adaptive link allocation (Fig. 2). We first constrain communication to a fixed exponential graph backbone with a small diameter, which provides scalable multi-hop information dissemination and preserves global reachability as the agent population grows. Within this constrained domain, we then learn a sparse and time-varying communication structure via a lightweight pairwise scoring network that adaptively selects task-relevant links conditioned on agent states. To ensure that the learned communication focuses on globally useful signals, auxiliary grounding objectives encourage messages to encode task-relevant information, guiding adaptive selection toward informative connections and stabilizing optimization. Together, these components enable scalable yet task-adaptive communication without redesigning the

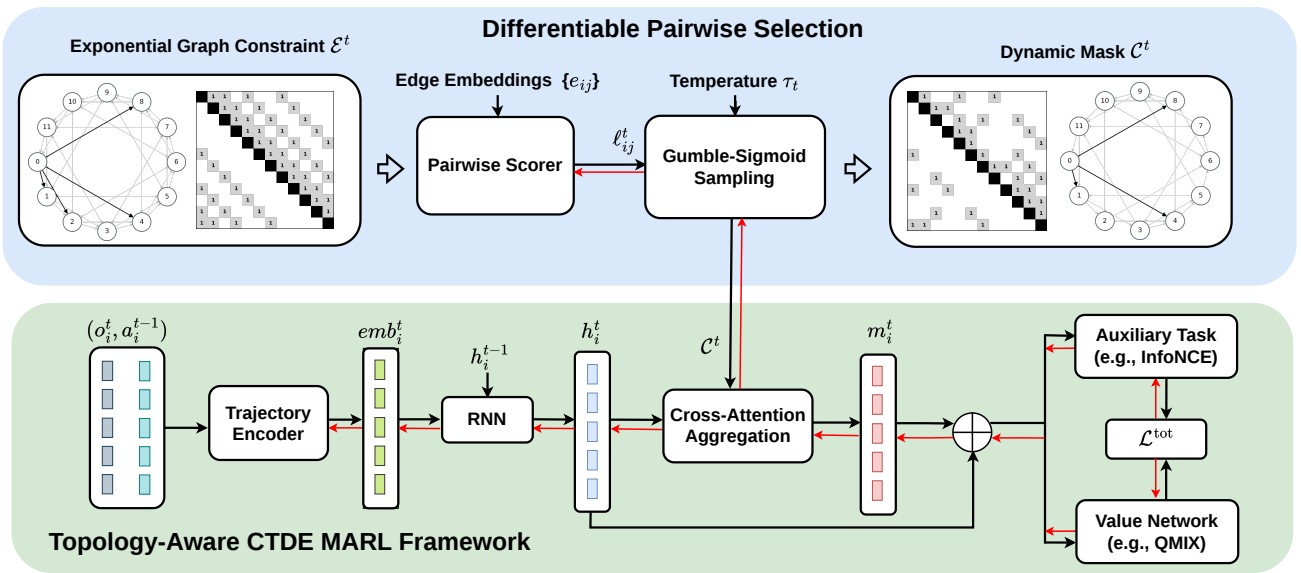

*Figure 2.* Overview of the SOPS architecture. The upper module learns a time-varying sparse mask over an exponential communication backbone via shared pairwise scoring and Gumbel-Sigmoid sampling. The lower module is a topology-aware CTDE MARL framework, where the selected mask drives cross-attention message aggregation and the fused states are used for value-based decision learning. An auxiliary objective regularizes message learning for more stable training. Red arrows denote backward gradient flow.

topology for different population sizes.

### 4.1. Exponential Graph Constraint

Exponential graphs are a family of sparse communication topologies that enable efficient information dissemination among nodes (or agents) with low communication overhead (Ying et al., 2021; Chen et al., 2021). They are particularly well-suited for decentralized and multi-agent systems due to their small diameter and near-linear scaling in communication cost. In an exponential graph, each node communicates with a fixed set of neighbors determined by powers of two. Specifically, each node $i$ (indexed from 0 to $N-1$) connects to nodes at distances $2^0, 2^1, \ldots, 2^{\lfloor \log_2(N-1) \rfloor}$ modulo $N$. This results in each node having a degree of $\lceil \log_2 N \rceil$. The corresponding adjacency matrix $\mathcal{E}_{ij}^t \in \{0,1\}^{N \times N}$ at timestep $t$ is defined as:

$$\mathcal{E}_{ij}^t = \begin{cases} 1 & \text{if } \log_2((j-i) \bmod N) \in \mathbb{Z} \text{ or } i=j, \\ 0 & \text{otherwise.} \end{cases} \quad (1)$$

This topology ensures that the graph diameter is $\lceil \log_2(N-1) \rceil$, meaning any two nodes can exchange information within logarithmic time steps. The communication cost is $N \cdot \lfloor \log_2(N-1) \rfloor$, which is efficient compared to fully connected graphs. We summarize the key scalability property of this backbone as follows.

**Proposition 4.1** (Sparsity and diameter). *The exponential backbone has $\lceil \log_2 N \rceil$ candidates per agent, $O(N \log N)$ total edges, and diameter $\lceil \log_2(N-1) \rceil$.*

This property implies logarithmic-hop reachability under a near-linear communication budget, making the exponential graph a scalable communication prior. To further highlight the advantages of static exponential connectivity, we conduct a comparative analysis against two representative alternatives: the Erdős–Rényi (ER) random graph and the Torus topology. The ER model is chosen as it has been shown to generate strong empirical performance and significantly more diverse connectivity patterns than Barabási-Albert or Watts-Strogatz models in multi-agent systems by Lou et al. (2024). In contrast, the Torus exemplifies a purely local, low-degree regular structure, which provides a natural worst-case comparison. As shown in Fig. 1, each heatmap cell records the average coverage fraction over multiple random sources (and graph resamples for ER). The results reveal a consistent ordering: the static exponential topology rapidly reaches near-complete coverage with a small $c$ and maintains scalability as $N$ grows, ER requires a larger $c$ and exhibits gradual degradation with scale, Torus remains the slowest, especially for large-scale systems. These findings confirm that the static exponential graph offers the most efficient and scalable backbone topology for communication in large-scale MARL.

### 4.2. Differentiable Task-Adaptive Selection

Optimizing a binary communication matrix $\mathcal{C}^t \in \{0,1\}^{N \times N}$ is difficult because it requires a differentiable mechanism that outputs discrete $0/1$ entries. A standard remedy is to treat $\mathcal{C}^t$ as a random matrix drawn from independent Bernoulli variables supported on the backbone, i.e.,

$\mathcal{C}_{ij}^t \sim \text{Ber}(\Theta_{ij}^t)$ with $\Theta^t \in [0,1]^{N \times N}$ and $\Theta^t$ masked by $\mathcal{E}^t$. To make this scalable, instead of optimizing $N^2$ free probabilities, we parameterize $\Theta^t$ via shared pairwise scoring on node embeddings only for backbone candidates. At step $t$, agent $i$'s candidate set is $\mathcal{N}_i^t = \{ j \mid \mathcal{E}_{ij}^t = 1 \}$. Since $|\mathcal{N}_i^t| = O(\log N)$, the topology-selection scorer evaluates only $O(N \log N)$ candidate pairs per step, rather than all $N^2$ agent pairs.

For each candidate pair $(i,j)$, we compute a lightweight score $\ell_{ij}^t = f(h_i^t \| h_j^t \| e_{ij})$ and convert it into a Bernoulli parameter $\theta_{ij}^t = \sigma(\ell_{ij}^t)$, where $e_{ij}$ is an edge-type embedding and $\sigma$ denotes the logistic function. Stacking these pairwise probabilities yields a proposal matrix $\sigma(\mathcal{G}(H^t))$, which is then masked by the backbone to obtain $\Theta^t = \sigma(\mathcal{G}(H^t)) \odot \mathcal{E}^t$, with $\odot$ representing element-wise multiplication. Conceptually, $\ell_{ij}^t$ is a sender-anchored topology-selection score for the directed edge $i \rightarrow j$: for each sender $i$, the scorer evaluates only backbone candidates $j \in \mathcal{N}_i^t$. In our CTDE implementation, these scores are batch-computed centrally on the learner from the hidden-state tensor $H^t$, which is already available during training; this does not introduce additional all-to-all communication among agents. The edge-type embedding $e_{ij}$ is a fixed feature of the backbone used by the scorer, not an extra transmitted message. Thus, SOPS adds only a lightweight topology-selection overhead bounded by the backbone degree, while task messages are transmitted only along selected edges.

Since gradients cannot pass through Bernoulli sampling directly, we adopt a Gumbel–sigmoid reparameterization with a straight-through estimator. We draw two i.i.d. Gumbel variables $g_{ij}^1, g_{ij}^2 \sim \text{Gumbel}(0,1)$ and use their difference to obtain logistic noise $\varepsilon = g_{ij}^1 - g_{ij}^2 \sim \text{Logistic}(0,1)$. Then we apply the reparameterization trick proposed by Maddison et al. (2017):

$$y_{ij}^t = \text{sigmoid}\left( \left( \log\left( \theta_{ij}^t/(1 - \theta_{ij}^t) \right) + \varepsilon \right) /\tau_t \right), \quad (2)$$

where $y_{ij}^t$ is the soft gate from the Gumbel relaxation and $\tau_t$ is the temperature at timestep t. In communication selection, deployment typically requires hard (0/1) links, so annealing is aligned with the final objective: begin with a higher temperature to obtain a smooth surrogate and sufficient exploration, then gradually reduce it so that the gates approach binary and stabilize (Jang et al., 2017). This makes an annealed schedule ($1.0 \rightarrow 0.3$) typically yield smoother optimization early and a more stable topology later. Accordingly, we set $\tau_t = \max(\tau_{\min}, \tau_{\max} - \beta t)$, where $\beta$ is the decay rate and $t$ is the training step. Then we take a hard threshold at 0.5 with a straight-through estimator, where $\mathbb{I}(\cdot)$ denotes the indicator function:

$$\mathcal{C}_{ij}^t = \mathbb{I}(y_{ij}^t > \tfrac{1}{2}) = y_{ij}^t + \text{stopgrad}(\mathbb{I}(y_{ij}^t > \tfrac{1}{2}) - y_{ij}^t). \quad (3)$$

Equivalently, since $\sigma$ is monotone, $\mathbb{I}(y_{ij}^t > \tfrac{1}{2}) = \mathbb{I}(\ell_{ij}^t + \varepsilon > 0)$, hence the sample follows $\mathcal{C}_{ij}^t \sim \text{Ber}(\theta_{ij}^t)$ and is inde-

pendent of $\tau_t$, the temperature only controls the smoothness of the surrogate and the gradient scale. At inference time, we use the deterministic rule $\mathcal{C}_{ij}^t = \mathbb{I}(\ell_{ij}^t > 0)$.

### 4.3. Message aggregation mechanism

At each time step $t$, agent $i$ first aggregates messages from its selected in-neighbors at the previous step, where each incoming message $m_j^{t-1}$ is gated by the binary communication indicator $\mathcal{C}_{ji}^t$ and the in-neighbor set is $\mathcal{N}_i^t$. Concretely, we form the aggregated message as $m_i^t = \text{Agg}(\{ \mathcal{C}_{ji}^t \cdot m_j^{t-1} : j \in \mathcal{N}_i^t \})$, and then fuse it with the current hidden state via $\hat{h}_i^t = \psi(h_i^t \| m_i^t)$. Because maintaining all messages across multiple timesteps is memory-inefficient, we implement Agg as a lightweight scaled cross-attention mechanism. The fused state $\hat{h}_i^t$ then conditions the action-value network under CTDE.

### 4.4. Training Objective and Auxiliary Task

While the structural backbone and adaptive selection determine which agents communicate, they do not specify what information should be encoded in the exchanged messages. Without explicit supervision on message semantics, the learned communication may degenerate into several pathological behaviors. Agents may converge to trivial equilibria by transmitting zero messages or uninformative noise, as no learning signal encourages informative exchange. Communication can also overfit to local observations, failing to aggregate multi-hop or multi-timestep information available through the backbone. Moreover, agents may develop task-irrelevant coordination patterns that correlate with reward by chance but do not reflect meaningful coordination on task-relevant features. These issues arise because the RL objective provides only a sparse, delayed reward signal, which is insufficient to guide the high-dimensional message space (Hu et al., 2024). Moreover, in our method, the learned discrete communication graph creates a non-stationary optimization landscape where the value of a particular message depends on which links are active.

To help messages carry task-relevant global information, we follow the practice of using lightweight auxiliary grounding tasks that encourage local messages to accumulate and reflect multi-hop, multi-timestep information useful for decision making (Oord et al., 2018). When the global state is available during training, we supervise a small predictor to recover the current global state from local messages, encouraging messages to carry globally useful content:

$$\mathcal{L}_{\text{pred}}^{\text{Aux}}(\theta, \phi) = \mathbb{E}_{(s^t, o^t) \sim \mathcal{D}}\big[\, s^t - q(m_i^t; \phi)^2 \big], \quad (4)$$

where $m_i^t$ is the message of agent $i$ at time $t$ (with $i$ sampled uniformly), and $q(\cdot; \phi)$ is a learnable predictor used only for grounding (discarded at test time).

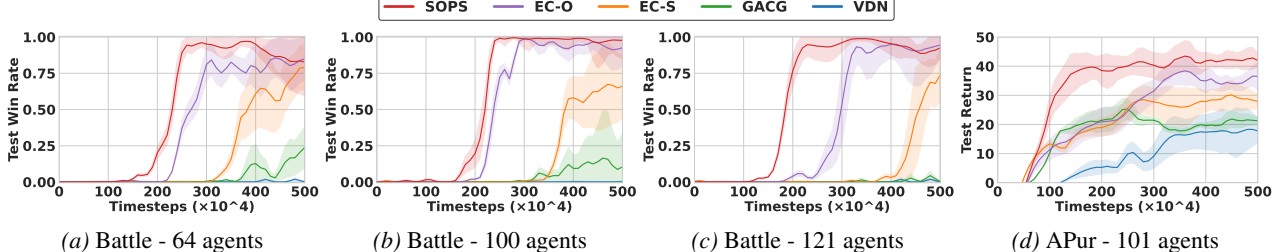

*(a)* Battle - 64 agents     *(b)* Battle - 100 agents     *(c)* Battle - 121 agents     *(d)* APur - 101 agents

*Figure 3.* Performance comparison of SOPS and baselines on Battle and AdversarialPursuit (APurs) scenarios. Win rate is the proportion of evaluation episodes won by the learning team. Return is the undiscounted cumulative reward per episode obtained by the learning (red) team during evaluation.

*Table 1.* Performance comparison of SOPS and baselines on three IMP scenarios with $N = 50$ and $N = 100$ agents. Results are mean $\pm$ std of normalized discounted rewards relative to heuristic baselines. The best-performing method is in **bold**, the second best is underlined.

| Scenario | VDN | GACG | EC-O | EC-S | SOPS |
|---|---|---|---|---|---|
| | | | $N = 50$ | | |
| Uncorrelated | 23.43 ($\pm$6.27) | 25.51 ($\pm$4.50) | 27.68 ($\pm$4.89) | 28.21 ($\pm$4.40) | **29.10 ($\pm$4.21)** |
| Correlated | 18.04 ($\pm$8.78) | 39.50 ($\pm$7.55) | 42.10 ($\pm$3.44) | 44.02 ($\pm$5.47) | **45.59 ($\pm$5.03)** |
| OWF | 61.32 ($\pm$2.23) | 63.49 ($\pm$1.86) | 64.82 ($\pm$1.66) | 63.70 ($\pm$1.63) | **65.46 ($\pm$2.55)** |
| | | | $N = 100$ | | |
| Uncorrelated | 8.10 ($\pm$7.73) | 23.04 ($\pm$12.93) | 28.62 ($\pm$4.66) | 27.19 ($\pm$10.24) | **29.21 ($\pm$5.90)** |
| Correlated | -54.80 ($\pm$100.44) | 12.74 ($\pm$25.83) | 18.85 ($\pm$19.36) | 21.62 ($\pm$20.35) | **23.55 ($\pm$19.94)** |
| OWF | 64.88 ($\pm$2.21) | 65.60 ($\pm$0.78) | 65.96 ($\pm$0.50) | 63.76 ($\pm$1.21) | **66.40 ($\pm$0.48)** |

Otherwise, when the global state is unavailable, we adopt a contrastive InfoNCE objective that treats same-timestep messages from different agents as positives and messages outside a backbone-diameter temporal window (or from other agents/timesteps) as negatives:

$$
\mathcal{L}_{\text{cont}}^{\text{Aux}}(\theta) =
$$
$$
- \, \mathbb{E}_{i,j,t,t'} \left[ \log \frac{\exp(g(m_i^t) \cdot g(m_j^t)/\kappa)}{\sum_{m \in \mathcal{M}} \exp(g(m_i^t) \cdot g(m)/\kappa)} \right], \quad (5)
$$

where $i \sim \text{Unif}\{1,\ldots,N\}$, $j \sim \text{Unif}\{1,\ldots,N : j \neq i\}$, $g(\cdot)$ is an $\ell_2$-normalizing projection, $\kappa > 0$ is the temperature, and the set of negatives $\mathcal{M} = \{ m_k^{t'} : k \in \{1,\ldots,N\}$, negative timestep $t' \notin [t - \text{diam}(\mathcal{G}^t), t + \text{diam}(\mathcal{G}^t)] \} \cup \{ m_j^t \}$, $|\mathcal{M}| = M+1$, with $M$ the number of negative pairs (a hyperparameter). In our experiments, we use the global-state prediction loss in Eq. (4) for MAgent and the contrastive loss in Eq. (5) for IMP. The total training loss is:

$$
\mathcal{L}^{tot}(\theta) = \mathcal{L}^{\text{TD}}(\theta) + \alpha \cdot \mathcal{L}^{\text{Aux}}(\cdot), \quad (6)
$$

where $\mathcal{L}^{\text{Aux}}$ is (4) or (5) depending on whether the global state is available during training and $\alpha$ is the hyperparameter that balances the auxiliary term. The TD loss $\mathcal{L}^{\text{TD}}$ is defined

according to Qmix (Samvelyan et al., 2019):

$$
\mathcal{L}^{\text{TD}}(\theta) =
$$
$$
\mathbb{E}_{(s^t,o^t,a^t,r^t) \sim \mathcal{D}} \left[ \left( y^{\text{tot}} - Q_{\text{tot}}(s^t, o^t, a^t; \theta) \right)^2 \right], \quad (7)
$$

where $y^{\text{tot}} = r^t + \gamma \max_a Q_{\text{tot}}(s^{t+1}, o^{t+1}, a; \theta^-)$, and $\theta^-$ denotes the parameters of a periodically updated target network, as commonly employed in DQN (Van Hasselt et al., 2016).

## 5. Experiments

In this section, we demonstrate through extensive experiments in large-scale settings that SOPS achieves higher normalized performance (return/win rate), faster convergence, and stronger zero-shot transfer capabilities compared to various baseline methods. And in ablations, we demonstrate the effectiveness of linear temperature annealing in Gumbel reparameterization for our method and show that it significantly outperforms alternative gradient estimation approaches in terms of training stability and final performance. Experiments were conducted on a server running Ubuntu 24.04 LTS, equipped with two Intel Xeon Platinum 8468 CPUs, 8 × NVIDIA A800 80GB PCIe GPUs, and 2 TiB of system memory. All SOPS experiments across scenarios were completed within 34 hours. The code is available at https://github.com/DistriAI/SOPS. Detailed hyperparameters for all experiments are provided in Appendix A.

*Table 2.* Plugging SOPS into different existing methods in the IMP scenario with 50 agents. Results are mean ± std of normalized discounted rewards relative to heuristic baselines. The best-performing method is in **bold**, the second best is underlined.

| Scenario | QMIX | QMIX + SOPS | QPLEX | QPLEX + SOPS | SHAQ | SHAQ + SOPS |
|---|---|---|---|---|---|---|
| Uncorrelated | 24.13 (±5.29) | **29.10 (±4.21)** | 20.98 (±6.24) | 26.35 (±3.26) | -7.82 (±9.28) | 15.46 (±5.94) |
| Correlated | 18.04 (±7.37) | 45.59 (±5.03) | 20.10 (±4.82) | **46.26 (±6.32)** | -14.88 (±19.52) | 32.41 (±10.90) |
| OWF | 61.38 (±4.26) | **65.46 (±2.55)** | 60.74 (±3.18) | 62.93 (±2.36) | 49.90 (±13.79) | 58.38 (±7.69) |

*Figure 4.* Performance comparison of zero-shot transfer performance on the MAgent environment. Subfigures (a) – (g) are under the Battle scenario, while (h) – (k) are under the AdversarialPursuit (APurs) scenario. Baseline GACG is excluded due to its network architecture being incompatible with this ability.

## 5.1. Setup

**Benchmark.** We experiment with two large-scale benchmarks for MARL: MAgent (Zheng et al., 2018; Terry et al., 2020) and Infrastructure Management Planning (IMP) from Leroy et al. (2023). Using these benchmarks, we conduct evaluations of SOPS and various baseline methods across 15 distinct scenarios, where the number of agents varies from 20 to 121. All experiments are averaged over three random seeds, and for the MAgent result plots, the shaded areas represent the 95% confidence interval. More additional details about the benchmarks are provided in Appendix B.

**Baselines.** We compare SOPS against four representative methods: (i) *VDN* (Sunehag et al., 2017): Traditional value decomposition method; no explicit inter-agent messaging ($K = 0$). (ii) *GACG* (Duan et al., 2024): Learns group assignments and performs sparse message passing within and across groups in an end-to-end manner. (iii) Two ExpoComm variants, *EC-S (static)* and *EC-O (one-peer)* (Li et al., 2025b): EC-S uses a static exponential topology as the communication backbone on a ring of $N$ agents. EC-O employs a single-neighbor exponential schedule. This variant retains the $\lceil \log_2(N-1) \rceil$ diameter over time while keeping the instantaneous edge count linear ($N$ active edges per step). To ensure fairness, all methods are implemented in the same codebase and trained with a shared configuration whenever possible.

## 5.2. Main Results

**MAgent.** Across both AdversarialPursuit and Battle, SOPS consistently learns faster and attains higher asymptotic performance than all baselines (Fig. 3). In all four settings with different agent populations, SOPS exhibits an early performance take-off and saturates near the task ceiling (returns or win rate), while EC-O is typically the runner-up but converges more slowly and with larger variance. EC-S improves more gradually and plateaus lower, suggesting that a denser static backbone is less sample-efficient at scale. GACG and VDN lag behind across populations, with VDN (no explicit communication) particularly struggling as the number of agents increases. To further analyze the statistical significance of the results, we compute the Area-under-the-curve (AUC) for these four scenarios. The results in Table 8 show that SOPS exhibits a clear advantage in cumulative performance. Complete descriptions are provided in Appendix C.

**IMP.** The quantitative summary on IMP (Tab. 1) reveals these trends: SOPS achieves the best mean normalized discounted reward in all tasks at both $N$=50 and $N$=100 ($N$ is the number of agents), with clear margins on *Correlated* and solid gains on *Uncorrelated* and *OWF*. EC-O/EC-S are competitive but consistently below SOPS, while VDN degrades sharply under correlation at $N$=100. Overall, SOPS maintains performance as $N$ increases, indicating superior scalability relative to baselines.

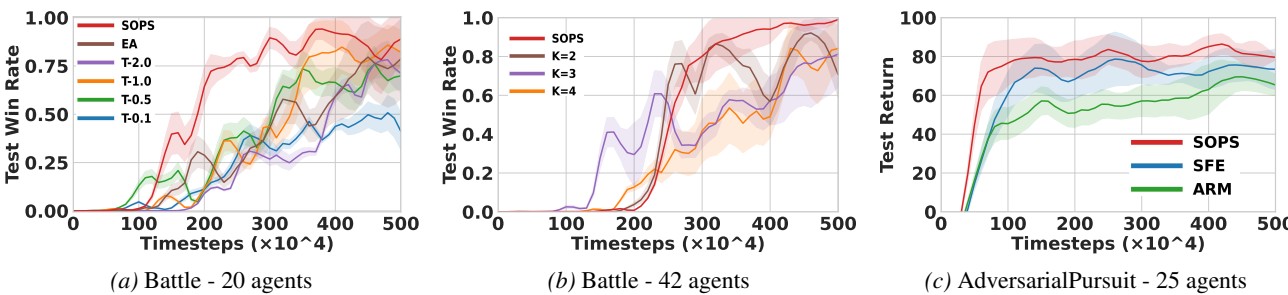

*(a)* Battle - 20 agents     *(b)* Battle - 42 agents     *(c)* AdversarialPursuit - 25 agents

*Figure 5.* Ablation studies on MAgent scenarios: (a) temperature (exponential annealing and $\tau = (0.1, 0.5, 1.0, 2.0)$), (b) edge-budget ($K = (2, 3, 4)$ communication links per agent), and (c) gradient-estimator variants (SFE, ARM).

**Plugability.** Our default implementation of SOPS is instantiated on top of QMIX. To verify that SOPS is not tied to a particular CTDE learner, we additionally evaluated it on the IMP benchmark with two additional value-decomposition methods, QPLEX (Wang et al., 2020) and SHAQ (Wang et al., 2022a), using the same training setup and hyperparameters. The results are summarized in Tab. 2. Across all three IMP scenarios, attaching SOPS to each learner consistently improves performance over its backbone, and the best or second-best method in every scenario is always a "+ SOPS" variant. This indicates that SOPS brings complementary benefits that persist under different and more expressive value-decomposition learners.

**Zero-shot transfer.** When transferring from smaller to larger populations without finetuning (Fig. 4), SOPS attains the highest win rates across all seven train→test pairs (e.g., $42 \to 81/100/121$). EC-O and EC-S retain certain transferability but drop more noticeably as the expansion factor grows; VDN performs poorly because of the lack of communication: trained at a small scale it learns only local heuristics, and transferring to larger populations leads to severe underfitting. GACG is excluded from the zero-shot transfer experiments because its fully-connected coordination graph and group structure are tied to a fixed agent set and are not directly reusable when the number or identities of agents change. These results demonstrate that SOPS preserves coordination under population scaling, aligning with its design goal of decoupling global reachability from task-adaptive sparse selection. To further understand this behavior, we conduct learned-graph diagnostics, which reveal that SOPS maintains sparse yet high-coverage communication structures under zero-shot transfer and that high-frequency edges remain largely stable across population changes. Detailed results are presented in Tab. 5 and Tab. 6 in Appendix C.2.

**Message Dropout.** To directly evaluate unreliable communication, we conduct test-time message dropout experiments on Battle-100 and AdversarialPursuit-101 after the communication schedules are determined, while keeping all learned policies unchanged. Each active message is independently dropped with probability $p \in \{0.1, 0.3, 0.5\}$. As shown

*Table 3.* Test-time message dropout on Battle-100. After each method determines its communication schedule, every active message is independently dropped with probability $p$. Values are mean $\pm$ std over three seeds.

| Method | $p = 0$ | $p = 0.1$ | $p = 0.3$ | $p = 0.5$ |
|--------|---------|-----------|-----------|-----------|
| SOPS | 0.98±0.04 | 0.95±0.05 | 0.88±0.10 | 0.74±0.14 |
| EC-O | 0.93±0.12 | 0.87±0.14 | 0.68±0.19 | 0.46±0.24 |
| EC-S | 0.67±0.22 | 0.64±0.24 | 0.54±0.25 | 0.39±0.26 |
| GACG | 0.12±0.10 | 0.10±0.07 | 0.07±0.09 | 0.04±0.06 |

in Tab. 3, all communication-based methods degrade as dropout increases, but SOPS remains the best-performing method and suffers the smallest overall degradation. The same trend holds on AdversarialPursuit-101, with results reported in Appendix C.3.

### 5.3. Ablations

**Gumbel Temperature.** We study the effect of the Gumbel temperature on edge sampling, as shown in Fig. 5a. Our default linear annealing (SOPS) delivers the earliest take-off and the highest final win rates at both 20- and 42-agent Battle scenarios. In contrast, exponential annealing (EA) cools too slowly early and too aggressively late, yielding delayed emergence of useful links and larger variance near convergence. Fixed temperatures underperform systematically: a high temperature (T-2.0) keeps gates overly soft and slows learning; a mid-value (T-1.0) improves but still lags; a lower value (T-0.5) learns faster but plateaus below SOPS; an overly low temperature (T-0.1) discretizes too early, hurting exploration and stability. These trends indicate that a smooth, schedule-driven hardening of edges is crucial for both sample efficiency and final performance, with linear cooling proving most robust across scales.

**Edge Budget.** Fig. 5b illustrates the benefit of learned neighbor selection over the exponential backbone with different edge budgets. In the 42-agent Battle setting, each agent has five candidate neighbors on the backbone. We remove the scorer and perform an ablation on the edge budget K, selecting edges based on Euclidean distance as the

static feature. As shown in the figure, SOPS exhibits rapid improvement after the warm-up phase and converges stably to a high win rate. In contrast, the static baselines with K=2 and K=3 exhibit pronounced late-stage oscillations, indicating insufficient connectivity for stable coordination. The K=4 variant is more stable but learns slowly and fails to match SOPS's final performance. These results indicate that while the exponential backbone already provides a strong inductive bias, the main additional gains in both final performance and stability come from the learned, adaptive neighbor selection.

**Gradient Estimator.** Fig. 5c compares the performance of our default reparameterized Gumbel sampling approach against two representative score-function gradient estimators: the Score Function Estimator (SFE, also known as RE-INFORCE) and the Augment-REINFORCE-Merge (ARM) estimator (Yin & Zhou, 2019). SFE provides an unbiased but high-variance gradient estimate that requires careful baseline subtraction for stable training. ARM, specifically designed for binary variables, leverages symmetry properties to construct a lower-variance estimator, offering better performance than SFE but still lacking the gradient stability of reparameterization methods. These results confirm that direct gradient flow through the sampling process, rather than score-function based estimation, is essential for achieving both high performance and training stability in discrete decision-making scenarios.

## 6. Conclusion

We present SOPS, a scalable communication scheme for cooperative MARL that addresses the scalability–adaptivity trade-off via a structured design. We first constrain communication to a fixed small-diameter exponential backbone to preserve rapid potential reachability at scale, and then learn a task-conditioned probabilistic subgraph on top of this domain through lightweight pairwise scoring, yielding sparse and time-varying link allocation without redesigning the topology across population sizes. We further incorporates message grounding objectives to encourage exchanged messages to encode globally task-relevant signals, preventing degenerate communication (e.g., uninformative messages or task-irrelevant protocols) and stabilizing the learning of sparse structures. Experiments show that SOPS achieves higher returns and faster convergence, while exhibiting zero-shot transfer to larger agent populations. With its pluggable compatibility with CTDE backbones, SOPS provides a practical path toward efficient, adaptable, and scalable communication in large-scale MARL.

## Impact Statement

This paper presents work whose goal is to advance the field of machine learning. There are many potential societal consequences of our work, none of which we feel must be specifically highlighted here.

## Ackowledgment

This work was supported in part by the National Natural Science Foundation of China (No. 62576213, 62472460), Guangdong Province Key Projects in Artificial Intelligence (Intelligent Robots) (No. 2025ZDZX3047), the Innovation Team Project of Guangdong Province of China (2024KCXTD017), and Shenzhen Loop Area Grant (FP202602).

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

## A. Hyperparameters

To ensure the fairness of comparison, we implement SOPS and all baselines within the same codebase, using shared hyperparameters except for those unique to specific methods. Following Li et al. (2025b), we use the default settings provided in the official implementations of MAgent (Zheng et al., 2018; Terry et al., 2020) and IMP (Leroy et al., 2023). For GACG (Duan et al., 2024), we use the officially recommended parameters: number of groups = 2 and trajectory length for group division = 10. For SOPS and two ExpoComm variants (Li et al., 2025b): loss coefficient $\alpha = 0.1$, loss temperature $\kappa = 0.07$ and number of negatives $M = 20$. Other common hyperparameters are given in Tab. 4.

*Table 4.* Common hyperparameters.

| Hyperparameter | MAgent | IMP |
| --- | --- | --- |
| Hidden sizes | 64 | 64 |
| Discount factor $\gamma$ | 0.99 | 0.95 |
| Batch size | 32 | 64 |
| Replay buffer size | 2000 | 2000 |
| Number of environment steps | $5 \times 10^6$ | $2 \times 10^6$ |
| Epsilon anneal steps | $5 \times 10^5$ | $5 \times 10^3$ |
| Test interval steps | $5 \times 10^4$ | $2.5 \times 10^4$ |
| Number of test episode | 100 | 100 |

## B. Benchmarks and Scenarios

**MAgent**   MAgent is a highly scalable many-agent reinforcement learning platform built on a large gridworld engine, supporting up to millions of agents on a single GPU through parameter sharing and agent ID embeddings (see Fig. 6). It provides flexible configuration for environments and agents, a reward description language enabling event-driven incentives, and an interactive renderer for real-time visualization. Agent actions are discrete — including move, turn, attack, or tag — and the underlying gridworld engine facilitates fast simulation of heterogeneous agent populations.

Two canonical scenarios within MAgent are AdversarialPursuit and Battle. In AdversarialPursuit, predators receive positive rewards upon successfully tagging prey, while prey incur penalties when tagged (a typical reward structure is defined in the platform's API). After training, predator agents typically develop local cooperative behaviors, forming dynamic enclosures to trap preys and accumulate cumulative rewards over successive timesteps.

In Battle, two large teams, each comprising hundreds of agents, compete on a shared map. Agents select from discrete actions (move, attack, idle) to cooperatively eliminate opposing team members. A team wins either by eliminating all enemy agents or by having more surviving agents at the end of the episode. Through self-play training, agents often evolve hybrid global–local strategies, such as coordinated encirclement or guerrilla-style skirmishing, reflecting emergent team-level coordination.

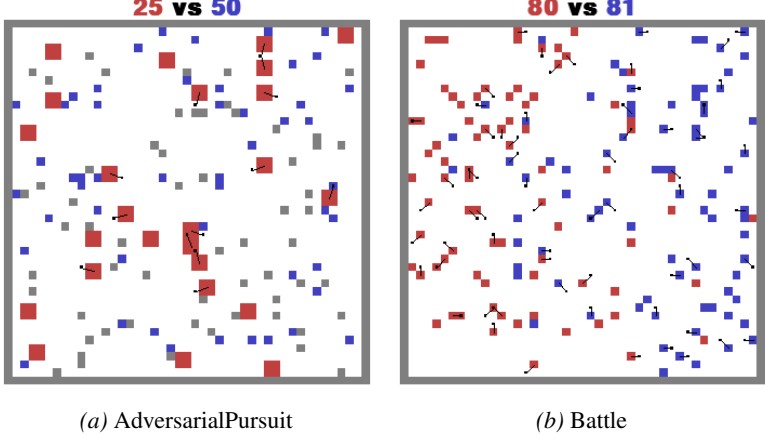

*(a)* AdversarialPursuit  *(b)* Battle

*Figure 6.* AdversarialPursuit and Battle scenarios in the MAgent environment. In both scenarios, red squares denote agents controlled by the trained MARL policy, while blue squares represent agents governed by pretrained IDQN policies (Rashid et al., 2020)

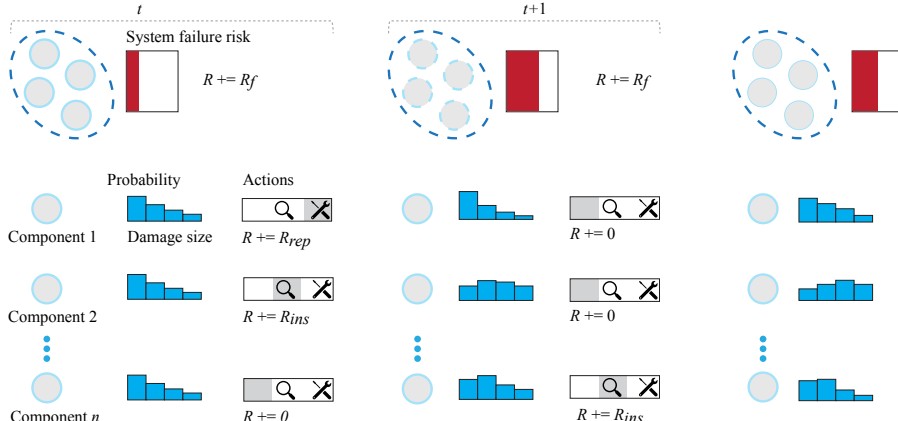

*Figure 7.* Infrastructure Management Planning (IMP) Environment. The system failure risk is modeled as a function of the probability distribution over each component's damage state. At every time step $t$, an agent—typically responsible for a single component—may choose to inspect or repair that component to regulate the failure risk. The IMP objective is to maximize the expected discounted return while trading off three (negative) reward terms: the system failure risk $R_f$, inspection costs $R_{ins}$, and repair costs $R_{rep}$. In the illustration, three components share the same damage probability at time $t$. If no action is taken, the damage probability follows a deterioration process.

**IMP** IMP is a cooperative multi-agent reinforcement learning benchmark for Infrastructure Management Planning, where each agent controls one system component and selects among three discrete actions: do nothing, inspect, or repair (see Fig. 7). Episodes are finite-horizon ($T = 20$–$30$), and the objective is to maximize the expected discounted return — balancing system failure risk against operation and maintenance (O&M) costs, with discount factor $\gamma = 0.95$. A configurable global campaign cost can be activated, imposing an additional timestep penalty for any inspection/repair action, thereby explicitly incentivizing coordinated scheduling across agents. The suite supports scaling to dozens or hundreds of agents via Gym/PettingZoo/PyMARL-compatible wrappers. Evaluation follows the authors' practice of normalizing returns relative to expert heuristic policies. Reward structure per step includes: (i) system failure risk $p_F^{\text{sys}}$ scaled by consequence, (ii) per-agent inspection/repair costs, and (iii) optional campaign cost — enabling evaluation of algorithms' ability to coordinate both spatially and temporally.

The benchmark includes three core scenarios: (1) Uncorrelated $k$-out-of-$n$: System fails if at least $(n - k + 1)$ components fail; damage states are represented by 30-bin probability vectors (last bin = failure), initialized independently per component. Observations concatenate normalized time with per-component damage distributions, testing coordination under independent deterioration. (2) Correlated $k$-out-of-$n$: Identical to above, but initial damage distributions are statistically correlated — inspecting one component reveals information about others. To mitigate partial observability, agents receive a shared correlation signal $\alpha_t$, updated from all inspection outcomes and appended to inputs, stressing cooperation under information coupling. (3) OWF (Offshore Wind Farm): Each turbine has three components (top/middle/mudline); mudline is unobservable/unrepairable, so two agents control each turbine (top + middle). Damage uses 60 bins, component models/costs vary by location, and turbines fail if any component fails; farm-level risk aggregates over turbines. This scenario emphasizes heterogeneous components and large-scale fleet management.

## C. Additional Details and Discussion

### C.1. For MAgent Benchmark

To complement the main-figure analyses, we additionally report the area-under-curve (AUC) metrics for the evaluation curves of Battle and AdversarialPursuit. Unlike final performance alone, AUC provides a holistic measure of learning progress by integrating both convergence speed and asymptotic performance over the entire training horizon. For the top-3 method (EC-S, EC-O, and SOPS) and each scenario, we compute AUC separately for all three random seeds and report the mean and standard deviation. Tab. 8 show that SOPS consistently achieves higher AUC across both benchmarks, indicating superior sample efficiency and overall training stability. This further confirms the advantages of SOPS observed in Fig. 3, even in cases where confidence intervals visually overlap due to near-saturation of the evaluation metrics.

*Table 5.* Graph sparsity and multi-hop coverage across zero-shot transfer settings. For each train→test pair, we report the average out-degree and the average fraction of reachable agent pairs within $h$ hops (coverage@$h$) measured at test time.

| Setting (train → test) | Avg. out-degree | Coverage@2-hop | Coverage@3-hop | Coverage@4-hop |
|---|---|---|---|---|
| Battle 20→64 | 3.21 | 0.34 | 0.61 | 0.88 |
| Battle 20→81 | 3.35 | 0.20 | 0.45 | 0.63 |
| Battle 42→81 | 4.32 | 0.39 | 0.68 | 0.90 |
| Battle 42→100 | 4.46 | 0.27 | 0.55 | 0.72 |
| Battle 42→121 | 4.58 | 0.18 | 0.40 | 0.53 |
| AdversarialPursuit 25→45 | 3.33 | 0.42 | 0.66 | 0.93 |
| AdversarialPursuit 25→61 | 3.49 | 0.35 | 0.57 | 0.72 |
| AdversarialPursuit 45→61 | 4.04 | 0.32 | 0.59 | 0.75 |
| AdversarialPursuit 45→101 | 4.35 | 0.24 | 0.46 | 0.62 |

*Table 6.* Edge persistence across zero-shot transfer settings. For each train→test pair, we report the Pearson correlation between edge activation frequencies and the Jaccard overlap between the top-$k\%$ most frequently used edges.

| Setting (train → test) | Pearson $\rho$ | Jaccard@5% | Jaccard@10% | Jaccard@20% |
|---|---|---|---|---|
| Battle 20→64 | 0.93 | 0.39 | 0.81 | 0.60 |
| Battle 20→81 | 0.85 | 0.44 | 0.76 | 0.55 |
| Battle 42→81 | 0.95 | 0.47 | 0.88 | 0.68 |
| Battle 42→100 | 0.88 | 0.35 | 0.82 | 0.59 |
| Battle 42→121 | 0.85 | 0.40 | 0.74 | 0.50 |
| AdversarialPursuit 25→45 | 0.92 | 0.45 | 0.78 | 0.65 |
| AdversarialPursuit 25→61 | 0.88 | 0.37 | 0.70 | 0.50 |
| AdversarialPursuit 45→61 | 0.95 | 0.42 | 0.85 | 0.77 |
| AdversarialPursuit 45→101 | 0.87 | 0.28 | 0.76 | 0.52 |

## C.2. Learned-Graph Diagnostics

To better understand why SOPS transfers well under population resizing, we instrument the learned sparse communication graph during test-time rollouts, both in-distribution and under zero-shot transfer, and report structural diagnostics in Tab. 5–6. Tab. 5 summarizes graph sparsity and multi-hop coverage. Across all train→test pairs, the average out-degree remains in a narrow range around 3–4.5, substantially smaller than the exponential candidate set size. This indicates that SOPS consistently maintains a sparse per-agent communication load even after resizing. The coverage metrics in Tab. 5 further show that this sparse graph still provides fast information propagation. In moderate resize regimes such as Battle $42 \rightarrow 81$ and AdversarialPursuit $25 \rightarrow 45$, coverage within 3–4 hops quickly approaches $0.9$ or higher, implying a small effective diameter and near-global reachability in just a few communication rounds. For more extreme enlargements (e.g., Battle $20 \rightarrow 100/121$ or AdversarialPursuit $45 \rightarrow 101$), coverage@3 and coverage@4 decrease but remain substantially above what would be expected from random sparse graphs, mirroring the larger but still moderate drop in zero-shot performance in Fig. 4.

Tab. 6 complements these structural statistics with an explicit measure of edge persistence across resize. For each train→test pair, we compute the activation frequency of every directed edge during evaluation and compare the resulting frequency matrices. The Pearson correlation between edge frequencies is consistently high ($0.79$–$0.95$), indicating that edges that are frequently used at the train population size tend to remain frequently used after resizing. Moreover, the Jaccard overlap of the top-$10\%$ most frequently used edges lies between $0.67$ and $0.88$ across all settings, showing that the majority of high-utility communication channels are reused rather than being replaced by entirely new links. Overlaps at $5\%$ are slightly lower due to the small set size and ranking noise, while overlaps at $20\%$ remain clearly above chance, suggesting that even a broader band of mid- to high-frequency edges is largely preserved.

*Table 7.* Test-time message dropout on AdversarialPursuit-101. After each method determines its communication schedule, every active message is independently dropped with probability $p$, while the learned policies remain unchanged. Values are mean $\pm$ std over three seeds.

| Method | $p = 0$ | $p = 0.1$ | $p = 0.3$ | $p = 0.5$ |
|--------|---------|-----------|-----------|-----------|
| SOPS | 41.39$\pm$1.42 | 39.96$\pm$1.82 | 36.58$\pm$2.93 | 31.72$\pm$4.16 |
| EC-O | 36.21$\pm$4.28 | 34.21$\pm$4.75 | 29.48$\pm$5.86 | 23.36$\pm$7.00 |
| EC-S | 27.81$\pm$2.43 | 27.01$\pm$2.87 | 24.16$\pm$3.51 | 19.31$\pm$4.69 |
| GACG | 21.28$\pm$1.33 | 20.36$\pm$1.54 | 17.52$\pm$2.20 | 13.28$\pm$3.14 |

*Table 8.* Area-under-the-curve (AUC) of evaluation curves for MAgent scenarios.

| Scenario | Battle-64 | Battle-100 | Battle-121 | AdvPursuit-101 |
|----------|-----------|------------|------------|----------------|
| SOPS | 0.51 ($\pm$0.05) | 0.57 ($\pm$0.01) | 0.60 ($\pm$0.05) | 31.65 ($\pm$3.34) |
| EC-O | 0.39 ($\pm$0.04) | 0.49 ($\pm$0.03) | 0.39 ($\pm$0.05) | 21.61 ($\pm$3.33) |
| EC-S | 0.18 ($\pm$0.06) | 0.16 ($\pm$0.05) | 0.08 ($\pm$0.01) | 18.82 ($\pm$2.95) |

### C.3. Robustness to Agent Failures and Topology Disruptions

Exponential graphs provide multi-scale, redundant connectivity—each agent links to $2^0, 2^1, \ldots$-offset neighbors—so the topology typically remains connected with near-logarithmic diameter even when some nodes or edges fail. Prior work similarly shows that exponential topologies preserve fast information mixing under random disruptions (Ying et al., 2021; Chen et al., 2021; Li et al., 2025b). On top of this backbone, SOPS does not depend on any specific edge: the pairwise scorer simply reallocates probability mass among the remaining candidates $N_i^t = \{j \mid E_{ij}^t = 1\}$, and cross-attention aggregation accumulates multi-hop information over time. Thus, the exponential skeleton offers inherent robustness, while task-adaptive sparse selection provides an additional mechanism to compensate for missing or unreliable links.

Although we do not perform targeted "agent failure" experiments, several of our existing results are consistent with robustness to topology perturbations. The broadcast-coverage study in Fig. 1 shows that exponential graphs maintain fast dissemination even when connectivity is effectively randomized or sparsified, outperforming Erdős–Rényi and Torus alternatives under matched budgets. Moreover, the zero-shot transfer experiments in Fig. 4 can be viewed as a strong form of structural perturbation: the agent population, and thus the communication graph, is resized between training and deployment without finetuning. SOPS preserves high coordination quality under these topology changes, suggesting that the learned communication scheme is not fragile to moderate structural variations.

We additionally report the AdversarialPursuit-101 message-dropout results in Tab. 7. In addition, our experimental suite primarily uses benchmarks with homogeneous agents (or weak heterogeneity), as is standard in large-scale MARL (e.g., MAgent, IMP). Algorithmically, SOPS itself does not rely on agent homogeneity: the exponential topology is defined over agent indices, and the pairwise scorer operates on learned embeddings that can naturally encode agent type, capability, or role. Nevertheless, we do not yet provide systematic evaluations in strongly heterogeneous, role-rich environments where highly targeted, type-aware communication may be especially beneficial. Extending SOPS with explicit type-aware edge features and assessing its performance in large-scale heterogeneous scenarios is a natural and important direction for future work.

### C.4. Limitations and Future Extensions

Our current analysis and experiments focus on settings where the exponential backbone can be instantiated over all active agents and remains largely intact during training and evaluation. We do not explicitly simulate adversarial or correlated failure patterns (e.g., removal of contiguous segments on the ring) or hard communication outages at scale. While the structural redundancy of exponential graphs and the task-adaptive selection in SOPS suggest a degree of inherent robustness, a systematic empirical study with backbone-edge dropout, synthetic agent failures, or failure-aware signals is left for future work.

## C.5. Visualization

In Fig. 8, we visualize the dynamic communication subgraphs induced by SOPS for a representative focal agent during zero-shot transfer on Battle scenarios. Each subfigure corresponds to a specific time step (t = 20, 25, 30, 35, 40, 45), capturing the evolution of communication patterns. Red and blue squares denote the SOPS-controlled team and opponents, respectively, with numerical labels above each panel indicating the remaining agent counts. The arrows highlight the focal agent's dynamically evolving communication links, revealing how SOPS maintains structured interaction topologies across varying population sizes and time steps.

In Fig. 9 we visualize the spatial behavior of the learned SOPS policies under zero-shot transfer on the Battle scenarios. Each row corresponds to a different train→test population resize (from top to bottom: 20→42, 20→64, 42→100, 42→121 agents), and each column shows a snapshot at different rollout times (t = 0, 25, 50, 75). Red squares denote the SOPS-controlled team and blue squares the opponent; the numbers above each panel indicate the remaining agents on each side. At t = 0 the transferred policies reproduce the characteristic lattice-like initial formations learned during training, now instantiated at larger population sizes. As time evolves, the agents consistently form coherent battle fronts and concentrated engagement zones rather than degenerating into random or fragmented patterns, and the qualitative engagement strategy (front-line formation, gradual encirclement, and clean-up of remaining opponents) is preserved across all resize settings. These visualizations illustrate that SOPS-induced communication graphs support robust zero-shot transfer, yielding stable and interpretable large-scale behaviors even when the population size at test time differs substantially from training.

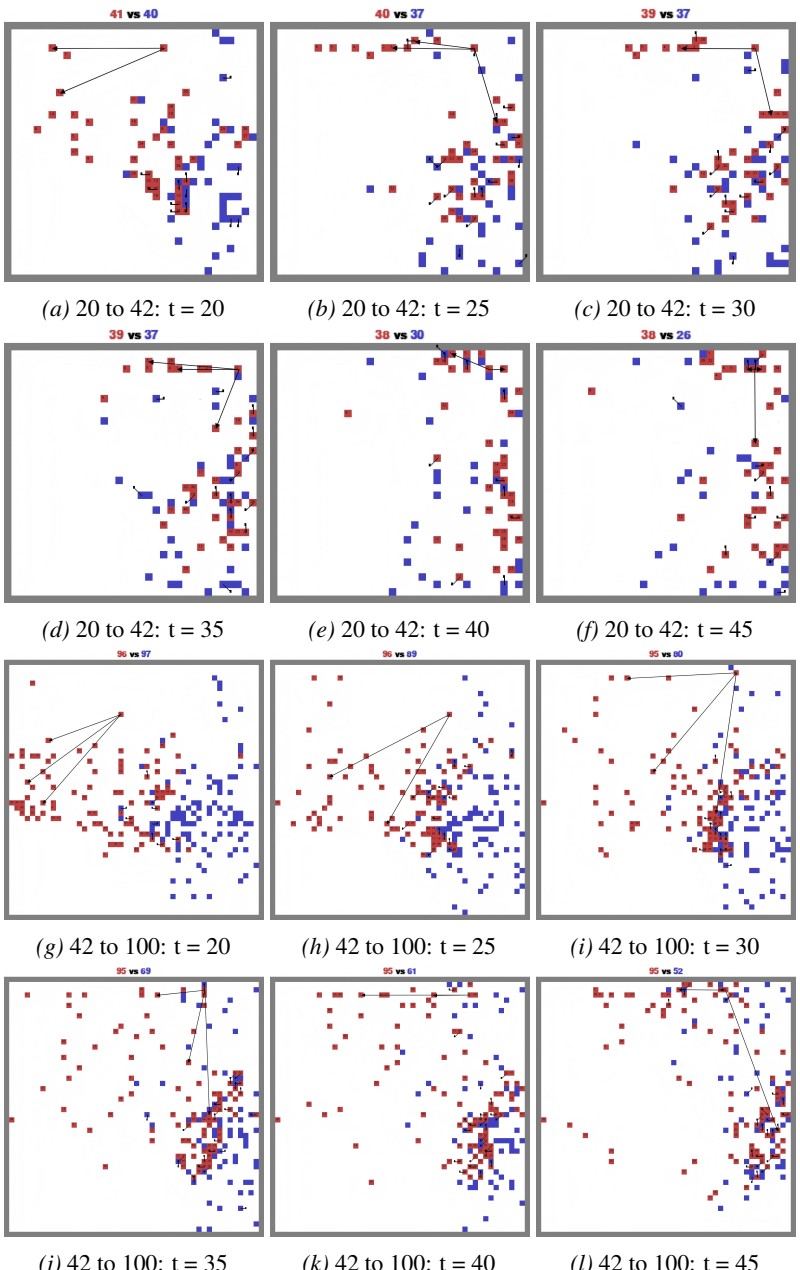

*(a)* 20 to 42: t = 20      *(b)* 20 to 42: t = 25      *(c)* 20 to 42: t = 30

*(d)* 20 to 42: t = 35      *(e)* 20 to 42: t = 40      *(f)* 20 to 42: t = 45

*(g)* 42 to 100: t = 20      *(h)* 42 to 100: t = 25      *(i)* 42 to 100: t = 30

*(j)* 42 to 100: t = 35      *(k)* 42 to 100: t = 40      *(l)* 42 to 100: t = 45

*Figure 8.* Visualization of SOPS-induced dynamic communication subgraphs under zero-shot transfer on Battle scenarios for a representative focal agent

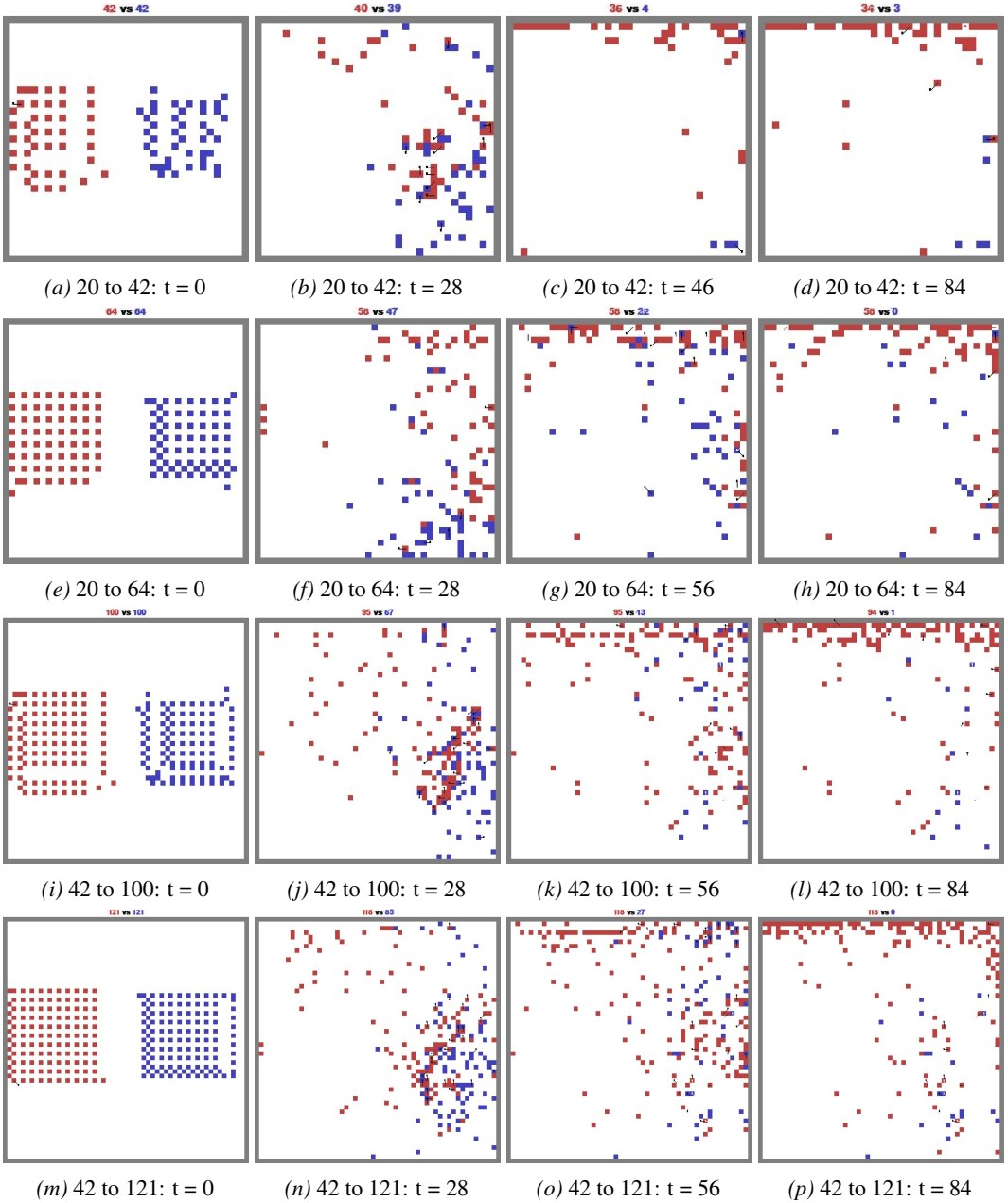

*(a)* 20 to 42: t = 0     *(b)* 20 to 42: t = 28     *(c)* 20 to 42: t = 46     *(d)* 20 to 42: t = 84

*(e)* 20 to 64: t = 0     *(f)* 20 to 64: t = 28     *(g)* 20 to 64: t = 56     *(h)* 20 to 64: t = 84

*(i)* 42 to 100: t = 0     *(j)* 42 to 100: t = 28     *(k)* 42 to 100: t = 56     *(l)* 42 to 100: t = 84

*(m)* 42 to 121: t = 0     *(n)* 42 to 121: t = 28     *(o)* 42 to 121: t = 56     *(p)* 42 to 121: t = 84

*Figure 9.* Visualization for zero-shot transfer performance on Battle scenarios.

