# OpenReview forum: "Sparse Topology-Aware Pairwise Scoring for Large-Scale Multi-Agent Reinforcement Learning"
_ICML.cc/2026/Conference — ICML 2026 regular_

### Official Review · Reviewer_jM4L · 2026-03-11

**Soundness:** 3
**Presentation:** 3
**Significance:** 4
**Originality:** 3
**Overall Recommendation:** 4
**Confidence:** 4

**Summary:**

This work aims to address the communication and scalability bottleneck in Large-Scale Multi-Agent Reinforcement Learning where traditional communication topologies suffer from increasing communication cost, causing severe bandwidth and memory bottlenecks. To address this, the authors propose a framework called Sparse tOpology Pairwise Scoring (SOPS). Instead of searching for a globally optimal communication graph from scratch, SOPS grounds its communication on a scalable exponential-graph backbone, which guarantees a small communication diameter. Over this backbone, SOPS learns a probabilistic subgraph distribution parameterized by a pairwise scoring network that takes agent states and edge-type embeddings as inputs. By leveraging the Gumbel-Sigmoid reparameterization trick, the model makes the discrete subgraph sampling differentiable. The evaluation results show that SOPS achieves higher rewards and faster convergence than state-of-the-art baselines (VDN, GACG, and ExpoComm variants) on cooperative benchmarks. Furthermore, the paper demonstrates that SOPS exhibits robust zero-shot transfer capabilities to larger-scale scenarios.

**Compliance With Llm Reviewing Policy:**

Affirmed.

**Final Justification:**

Thank you for addressing my comments and concerns in the rebuttal. Given also the other reviews and responses, I have raised my score to 'weak accept'.

**Key Questions For Authors:**

1) Does the need of agents’ internal representation and edge embeddings for the calculation of the lightweight score $l^t_{ij}$ indicate a centralized communication of agents ? Doesn’t this increase the required bandwidth?
2) Which approach is used in your experiments? The one regarding equation 4 or that of equation 5?

**Limitations:**

They don’t discuss the evaluation limitations which are about the lack of continuous action spaces. In modern MARL literature, this is an important limitation.

**Strengths And Weaknesses:**

Strengths:
From a technical perspective, the framework is well designed since grounding the dynamic graph search space within an exponential graph backbone is a reasonable architectural choice that ensures the communication diameter remains logarithmically small as the agent population grows. The use of Gumbel-Sigmoid reparameterization is mathematically sound and allows backpropagation through discrete graph sampling steps.
The paper is well-structured, starting with motivation, then continuing with related work on MARL cooperation and communication, and providing the required preliminaries of Dec-POMDP and graph communication modelling. The authors present the exponential graph communication topology which is required to understand later the adaptive mechanism of selecting the neighboring agents with which to communicate. They present how they ground messages to encapsulate useful information and enhance coordination among the agents. Finally they provide the evaluation results and the ablation study.
The results on the main part as well as on Appendix C1 indicate a significant improvement over the selected baselines, addressing in some extent the problem of scaling MARL to massive populations. Reducing the cost of message passing while maintaining strong cooperation is critical for real-world applications. Furthermore, the zero-shot transferability to larger populations, as shown by the experimental results on MAgent tasks, significantly increases the usefulness of this work, enabling new directions for training on small scales and deploying on large scales.
Regarding the originality of this work, while sparse graphs and Gumbel-Softmax routing have been explored in prior MARL literature, SOPS offers a novel combination of existing ideas. Specifically, bounding the dynamic pairwise scoring mechanism within a fixed, small-diameter exponential graph is a creative approach that distinguishes SOPS from purely dense or local interaction approaches.

Weaknesses:
In Figure 2 as well as in Section 4.2, it seems that the lightweight scores $l^t_{ij}$ are constructed using the internal representations of each agent ($h^t_i$, and $h^t_j$) and their edge embeddings. Does this require centralized communication of all the agents at least in each neighborhood? If yes, doesn’t this increase the requiring bandwidth? This should be clearly explained.
In Section 4.3, authors mention two different ways (Equation 4 and 5) to encourage messages to carry globally useful content, depending on whether global state is unavailable or not. Which of these approaches are used in the experiments? This must be explicitly stated.
Furthermore, in real-world scenarios some messages may be randomly dropped, so experiments that show the robustness of the framework in increasing message droppage scenarios should be provided.
Finally, while SOPS shows the best performance in the selected benchmarks, the evaluation is limited to discrete action spaces. On the other hand, SOTA MARL algorithms are mostly evaluated in continuous action spaces with increased task complexity, so definitely authors have to evaluate SOPS in more benchmarks, and also include algorithms that have attention-based communication mechanisms.

---

> ### Author Rebuttal · Authors · 2026-03-31
>
> We sincerely thank the reviewer for the positive assessment of our work and the constructive feedback. We address each concern below.
>
> **W1 & Q1: Communication Overhead of Pairwise Scoring**
>
> Thank you for raising this point. In SOPS, $\ell^t_{ij}=f(h_i^t,h_j^t,e_{ij})$ is computed **only on candidate edges of the exponential backbone**, not on all agent pairs. Thus, each agent considers only O(log N) candidates, and the total scoring cost is O(Nlog N) rather than O(N^2).
>
> Under CTDE, we batch-compute these scores centrally on the learner for efficiency, since the hidden states are already available there during training. This does **not** imply additional all-to-all communication among agents. Moreover, $e_{ij}$ is a fixed edge-type feature of the backbone, not an extra transmitted message. Therefore, **SOPS adds only a lightweight topology selection overhead, bounded by the backbone degree.**
>
> We will clarify this in the revision by explicitly distinguishing topology-selection signals from task messages.
>
> **W2 & Q2: Auxiliary Objective Used in Experiments**
>
> Thank you for pointing out this omission. In our experiments, we use **Eq. (4) on MAgent and Eq. (5) on IMP**. The reason is that MAgent provides a usable global state during training. In IMP, the global state is essentially a concatenation of observations, which doesn't work as a grounding target, so we switched to the contrastive objective in Eq. (5). We will add this clarification to the revised manuscript.
>
> **W3: Robustness to Message Dropout**
>
> We agree that robustness under unreliable communication should be evaluated explicitly. While the original submission only provided indirect evidence through the redundancy of the exponential backbone and the zero-shot transfer results under population resizing (Appendix C.2), we did not directly test random message loss. To address this, we additionally conducted **test-time message dropout experiments on Battle-100 and AdversarialPursuit-101 scenarios**, where after each method determines its communication schedule, each active message is independently dropped with a fixed probability p $\in$ {0.1, 0.3, 0.5} while keeping the learned policies unchanged. The results show that all communication-based methods degrade as dropout increases, but **SOPS remains the best overall and degrades the least, directly supporting its robustness to random message loss.**
>
> Dropout Battle-100
>
> ||0|0.1|0.3|0.5|
> |---|---|---|---|---|
> |SOPS|0.98±0.04|0.95±0.05|0.88±0.10|0.74±0.14|
> |EC-O|0.93±0.12|0.87±0.14|0.68±0.19|0.46±0.24|
> |EC-S|0.67±0.22|0.64±0.24|0.54±0.25|0.39±0.26|
> |GACG|0.12±0.10|0.10±0.07|0.07±0.09|0.04±0.06|
>
> Dropout AdversarialPursuit-101
>
> ||0|0.1|0.3|0.5|
> |---|---|---|---|---|
> |SOPS|41.39±1.42|39.96±1.82|36.58±2.93|31.72±4.16|
> |EC-O|36.21±4.28|34.21±4.75|29.48±5.86|23.36±7.00|
> |EC-S|27.81±2.43|27.01±2.87|24.16±3.51|19.31±4.69|
> |GACG|21.28±1.33|20.36±1.54|17.52±2.20|13.28±3.14|
>
> **W4: Evaluation on Continuous Action Spaces and Attention Baselines**
>
> Regarding the concern about discrete-only evaluation, we would like to clarify that SOPS operates at the communication level — it learns a sparse graph and produces fused states that can be passed to any policy head (discrete Q-network or continuous actor-critic), making it orthogonal to the action-selection mechanism. To empirically verify this, following the continuous-control MPE setting in [1], we evaluated SOPS, EC-O, and EC-S on continuous-action Spread with enlarged populations (20 and 30 agents) and restricted local observations using MAPPO. **SOPS achieves the best average episode reward, with its advantage growing at larger scales, confirming that topology-aware sparse communication benefits persist in continuous action spaces.**
>
> MPE Spread (Continuous)
>
> ||20 Agents|30 Agents|
> |---|---|---|
> |SOPS|-147.31±8.52|-202.47±7.12|
> |EC-O|-153.28±6.45|-239.63±8.07|
> |EC-S|-171.05±9.31|-315.90±10.22|
> |MAPPO|-211.93±5.26|-448±13.32|
>
> Regarding attention-based baselines, GACG already uses attention in its group-wise communication. We further added CACOM [2], a dedicated attention-based method on IMP, and it remains clearly below SOPS across all scenarios. **This suggests that attention alone is insufficient at scale**, whereas SOPS benefits from coupling topology-aware candidate restriction with adaptive link selection.
>
> IMP
>
> ||Uncorrelated-50|Uncorrelated-100|Correlated-50|Correlated-100|OWF-50|OWF-100|
> |---|---|---|---|---|---|---|
> |SOPS|29.10±4.21|29.21±5.90|45.59±5.03|23.55±19.94|65.46±2.55|66.40±0.48|
> |CACOM|8.54±8.71|-3.64±7.94|19.82±8.47|-10.75±27.82|55.86±4.79|61.37±1.80|
>
> [1] Zhong, Y., et al., 2024. Heterogeneous-agent reinforcement learning. Journal of Machine Learning Research.
>
> [2] Li, X., et al., 2024. Context-aware Communication for Multi-agent Reinforcement Learning. International Conference on Autonomous Agents and Multiagent Systems.

---

### Official Review · Reviewer_Ptns · 2026-03-12

**Soundness:** 3
**Presentation:** 3
**Significance:** 3
**Originality:** 3
**Overall Recommendation:** 4
**Confidence:** 3

**Summary:**

Many multi-agent reinforcement learning (MARL) systems allow agents to communicate, Most communication architectures require pairwise interactions between all agents.This results inO(N^2) communication complexity.As the number of agents grows  this becomes computationally infeasible.
This paper introduces Sparse Topology-Aware Pairwise Scoring (SOPS), a novel and well-structured approach to multi-agent communication that effectively balances connectivity and efficiency. The method's strength lies in its three-stage design: a clever Topology Constraint that pre-defines a sparse, structured communication graph for logarithmic information spread; a Pairwise Scoring mechanism using a neural network to estimate communication utility; and Sparse Edge Sampling to dynamically select only the most critical links at each timestep. Training with QMIX and an auxiliary InfoNCE loss seems robust, and the experimental results demonstrate that SOPS scales significantly better than dense communication models, maintaining strong performance even with large agent populations, and successfully learning task-effective, sparse communication patterns.

**Compliance With Llm Reviewing Policy:**

Affirmed.

**Key Questions For Authors:**

The theoretical complexity suggests scalability, but empirical evidence is limited.Testing with over thousand+ agents would strengthen the claims.

When agents operate in a physical environment, is the communication structure influenced by s distance and of their interactions?

**Limitations:**

Yes

**Strengths And Weaknesses:**

**Strengths**

Addresses a real scalability problem:

The paper targets the O(N²) communication bottleneck, which is a major limitation in MARL systems.

The proposed solution reduces complexity to O(Nlog⁡N) and this is a meaningful improvement.

Clever combination of ideas combining structured topology, learned pairwise routing and differential edge sampling which is practical and well designed

Consistent performance gains across benchmarks and ablation studies are meaningful

**Weakness**

Limited experiments scale, does it scale to 1000 agents, Whats the scale where it becomes meaningful?

How does the topology work without fixed indexing ex. spatial proximity

Gumbel sampling introduced variance and instability, are there insights on implications of this on training?

---

> ### Author Rebuttal · Authors · 2026-03-31
>
> We sincerely thank the reviewer for the positive assessment of our work and the constructive feedback. We address each concern below.
>
> **W1 & Q1: Experiment scale and evidence beyond 121 agents.**
>
> We appreciate these concerns. First, we note that our topology-level analysis in Fig. 1 already extends to N = 1024 agents, demonstrating that the exponential backbone maintains near-complete broadcast coverage within O(log N) rounds at this scale. **This provides structural evidence that the communication substrate underlying SOPS scales gracefully to thousand-agent regimes.**
>
> On the empirical side, **our experiments (up to 121 agents on MAgent and 100 agents on IMP) represent the largest scales evaluated among existing communication-focused MARL methods on these benchmarks.** Recent works addressing scalable MARL communication — including ExpoComm, GTDE [1] and MDPO [2] — similarly conduct experiments at scales of around one hundred agents, as current MARL benchmarks and training budgets make reliable evaluation at 1000+ agents challenging. We will also clarify this scope in the revised paper.
>
> Additionally, the zero-shot transfer experiments (Fig. 4) provide indirect evidence of scalability: policies trained at smaller populations (e.g., 20 or 42 agents) transfer effectively to substantially larger ones (up to 121 agents, a scale-up of 3-6 times). And the learned-graph diagnostics (Tabs 5–6) confirm that the communication structure remains sparse and well-connected after resizing. This suggests that SOPS's design principles do not break down as population size grows.
>
> [1] Li, M., et al., 2025. GTDE: Grouped training with decentralized execution for multi-agent actor-critic. Proceedings of the AAAI Conference on Artificial Intelligence, 39, pp.18368-18376.
>
> [2] Ma, C., et al., 2024. Efficient and scalable reinforcement learning for large-scale network control. Nature Machine Intelligence, 6(9), pp.1006-1020.
>
> **W2 & Q2: Topology without fixed spatial indexing and influence of spatial distance.**
>
> We thank the reviewer for these insightful questions. **SOPS does not require a spatially meaningful or fixed indexing.** The exponential topology is defined over any consistent agent ordering and serves only as a sparse candidate backbone with logarithmic degree and small diameter; it is not intended to encode physical proximity itself. The actual communication links are selected by the learned pairwise scorer based on agents’ hidden states and edge embeddings. Therefore, in physical environments, spatial distance and inter-agent interactions can influence communication to the extent that they are reflected in the observations and are relevant to the task, but they are not imposed as hard constraints by the topology itself. This is also consistent with our edge-budget ablation (Fig. 5b): replacing the learned scorer with a static Euclidean-distance heuristic leads to worse performance, **indicating that effective communication is governed by task relevance rather than spatial proximity alone.**
>
> **W3: Variance and instability from Gumbel sampling.**
>
> We agree this is an important practical consideration, and we have specifically investigated it through two ablation studies:
>
> **(1) Temperature annealing (Fig. 5a):** We compare our default linear annealing schedule with exponential annealing and four fixed temperatures (τ = 0.1, 0.5, 1.0, 2.0). Linear annealing achieves the earliest performance take-off and highest final win rate. High fixed temperatures (τ = 2.0) keep the gates overly soft, introducing excessive variance and slowing learning; very low temperatures (τ = 0.1) discretize too early, harming exploration. Linear annealing smoothly transitions from high-variance exploration to low-variance exploitation, providing the best trade-off.
>
> **(2) Gradient estimator comparison (Fig. 5c):** We compare Gumbel-Sigmoid reparameterization with two score-function estimators — SFE (REINFORCE) and ARM. Gumbel-Sigmoid substantially outperforms both in terms of training stability and final return. This confirms that the low-variance gradient flow enabled by Gumbel reparameterization is essential for stable training of discrete communication graphs.
>
> Together, these results show that while Gumbel sampling does introduce stochasticity, **the combination of reparameterized gradients and a well-tuned annealing schedule effectively controls variance and yields stable, high-performing training.**

---

> > ### Author Rebuttal · Reviewer_Ptns · 2026-04-05
> >
> > Thank you the response

---

> > > ### Author Response · Authors · 2026-04-07
> > >
> > > We sincerely thank you for acknowledging that the concerns have been fully resolved. We also appreciate your earlier comments on experimental scale, which motivated us to further strengthen the empirical evidence on **scalability** during the discussion period.
> > >
> > > **a) Resource scalability**. We measured the GPU memory consumption of all methods on the IMP benchmark with N=100, 200, 400, and 800 agents:
> > >
> > > |  | Correlated-100 | Correlated-200 | Correlated-400 | Correlated-800 |
> > > | --- | --- | --- | --- | --- |
> > > | SOPS | 5370 MB | 10468 MB | 26302 MB | 66742 MB |
> > > | EC-S | 5124 MB | 9670 MB | 23242 MB | 60938 MB |
> > > | EC-O | 4988 MB | 9355 MB | 20140 MB | 55832 MB |
> > > | GACG | 10210 MB | 22480 MB | 54175 MB | OOM |
> > >
> > > These results show that SOPS incurs only a modest memory overhead over the static-topology baselines due to its learned pairwise scorer, while remaining substantially more efficient than the fully connected GACG baseline, which runs out of memory at N=800. This additional cost enables task-adaptive link selection. **These results show that training at the scale of close to 1,000 agents can be accommodated on modern GPUs.**
> > >
> > > **b) Performance scalability**. We also added new IMP performance results at N=200 and N=400 (average return ± std; percentage indicates relative improvement over GACG):
> > >
> > > |  | Correlated-200 ↓ | Correlated-400 ↓ |
> > > | --- | --- | --- |
> > > | SOPS | -191.25±27.54 (+32.53%) | -259.44±31.97 (+30.04%) |
> > > | EC-S | -231.24±34.11 (+18.42%) | -294.57±42.28 (+20.57%) |
> > > | EC-O | -246.86±40.48 (+12.91%) | -318.65±35.04 (+14.07%) |
> > > | GACG | -283.42±57.30 | -370.82±55.49 |
> > >
> > > These additional results provide further evidence that SOPS remains effective at larger population sizes while retaining clear performance advantages. Together with the existing results, they offer a more complete empirical basis for our scalability claims. We hope these additional results help further address your concerns on scalability and will be taken into consideration in the final evaluation.

---

### Official Review · Reviewer_4mGF · 2026-03-13

**Soundness:** 2
**Presentation:** 3
**Significance:** 2
**Originality:** 2
**Overall Recommendation:** 4
**Confidence:** 3

**Summary:**

This paper proposes a scalable communication scheme (SOPS) to address Large-Scale MARL problem. The core of this method lies in the introduction of a technique termed "Exponential Graph Constraint." However, I observe that this concept bears significant similarity to the "exponential decay property" assumption found in existing research on Scalable MARL. Since this paper fails to compare its approach with or discuss its relationship to such methods, concerns arise regarding the method's advancement and novelty. In particular, compared to the SOPS algorithm proposed in this work, these existing Scalable MARL approaches offer a distinct advantage: they do not rely on centralized training.

**Compliance With Llm Reviewing Policy:**

Affirmed.

**Final Justification:**

The proposed SOPS method demonstrates a certain degree of novelty, and the experimental results support its conclusions. The authors' rebuttal has also addressed most of my concerns. However, I remain skeptical about research on large-scale reinforcement learning training under a centralized paradigm. In particular, compared to methods under the decentralized paradigm, such approaches seem to have inherent limitations when facing large-scale training.

**Key Questions For Authors:**

1. This paper proposes the SOPS algorithm, aiming to address scalability and adaptability in MARL for large-scale multi-agent scenarios, and presents comparisons with classic MARL algorithms under the CTDE framework. However, given the existence of numerous existing works on large-scale reinforcement learning (e.g., MDPO [1]), I suggest the authors include comparative results with these methods.
2. The paper employs "Exponential Graph Constraint" technology. Is this similar to the "Exponential decay property" assumption found in algorithms like SAC [2] and Scal-MAPPO-L [3]? If not, what are the specific differences?
3. Is the proposed method applicable in scenarios outside the CTDE (Centralized Training with Decentralized Execution) framework?

## Reference
[1] Ma, C., Li, A., Du, Y., Dong, H. and Yang, Y., 2024. Efficient and scalable reinforcement learning for large-scale network control. Nature Machine Intelligence, 6(9), pp.1006-1020.

[2] Ying, D., Zhang, Y., Ding, Y., Koppel, A. and Lavaei, J., 2023. Scalable primal-dual actor-critic method for safe multi-agent rl with general utilities. Advances in Neural Information Processing Systems, 36, pp.36524-36539.

[3] Zhang, L., Li, L., Wei, W., Song, H., Yang, Y. and Liang, J., 2024. Scalable constrained policy optimization for safe multi-agent reinforcement learning. Advances in Neural Information Processing Systems, 37, pp.138698-138730.

**Limitations:**

Please refer to the “Strengths And Weakness” section for related questions.

**Strengths And Weaknesses:**

## Strengths
1. This paper clearly articulates the research problem and its proposed solution.
2. The writing quality of the paper is good, and its structure is reasonable.

## Weaknesses
1. In terms of methodological characteristics, the approach proposed in this paper lacks novelty. Furthermore, the paper fails to adequately articulate its superiority over existing methods.
2. I am skeptical about the significance of addressing communication issues specifically within the CTDE framework. Compared to existing Scalable MARL methods, the scalability of the proposed approach does not appear to offer a distinct advantage.
3. The paper validates the effectiveness of the proposed method by combining it with classic multi-agent algorithms such as QMIX, QPLEX, and SHAQ. However, why were these not integrated with the latest methods under the CTDE framework? To my knowledge, there are numerous such recent approaches that would serve as more relevant baselines.
4. Given the adoption of the "Exponential Graph Constraint," the algorithm may possess specific theoretical properties. The authors should provide a theoretical analysis to substantiate these characteristics.

---

> ### Author Rebuttal · Authors · 2026-03-31
>
> We sincerely thank the reviewer for the thorough and detailed review. We address each point below.
>
> **W1: Novelty and Differentiation from Existing Methods**
>
> We clarify our contribution. Scalable communication under CTDE remains an open challenge: learned methods (e.g., GACG) often suffer O(N^2) bandwidth or fixed population size, while topology-based methods (e.g., ExpoComm) provide bounded bandwidth but cannot adapt links to task dynamics. SOPS is the first to decouple these concerns — an exponential backbone ensures near-linear scalability, while learned pairwise scoring selects task-conditioned sparse subgraphs atop it. **No prior method simultaneously achieves bounded bandwidth, task-adaptive selection, and zero-shot transferability.**
>
> Our technical components are tightly coupled: the backbone bounds the search space, Gumbel-Sigmoid reparameterization enables gradient flow through discrete decisions, and auxiliary grounding stabilizes the non-stationary optimization landscape. Consistent gains across all scenarios validate this integrated design (Figs. 3–5, Tabs. 1–2).
>
> **W2 & Q1: Significance of CTDE communication and comparison with MDPO**
>
> CTDE is the most common and practical training/execution setting for cooperative Dec-POMDPs. However, **communication bottlenecks persist under CTDE**: when the critic/value network must handle large-scale agent interactions during centralized training, dense or all-to-all message passing quickly exceeds available bandwidth.
>
> Regarding MDPO, the two methods address complementary problems. MDPO focuses on sample-efficient policy optimization via localized world models in systems with pre-defined physical topologies (e.g., CACC, Power Grids). SOPS addresses a different question: given N agents without an inherent interaction graph, how to construct and dynamically select sparse communication links preserving global reachability and adaptivity. **A direct empirical comparison is unfortunately not straightforward**: MDPO's benchmarks assume fixed physical adjacency, which eliminates the topology design problem SOPS targets, while MAgent/IMP lack the factored transition structures that MDPO's localized world models require. That said, **the two approaches are naturally complementary: one could combine SOPS's communication topology with MDPO's model-based optimization**, which we view as a promising future direction. We will add a positioning discussion in the revision to clarify this relationship.
>
> **W3: Choice of Baselines for Pluggability Evaluation**
>
> We clarify that **the experiments in Tab. 2 (QMIX, QPLEX, SHAQ) are designed to demonstrate SOPS's pluggability** — i.e., that our communication module provides consistent gains when attached to diverse value decomposition backbones, rather than being tied to a specific learner. The choice of these three methods reflects architectural diversity, which is the relevant axis for a pluggability study. Since SOPS operates at the communication layer and is agnostic to the underlying value decomposition method, we expect this pluggability to extend naturally to newer CTDE learners as well.
>
> **W4 & Q2: Relation to Exponential Decay and Theoretical Analysis**
>
> The exponential decay property in SAC and Scal-MAPPO-L is an assumption on environment dynamics: agent j's influence on agent i's transition decays exponentially with graph distance. Our exponential graph constraint is an engineered sparse communication topology. **The two are complementary: if influence decays exponentially, connecting at exponentially increasing distances efficiently captures long-range interactions within a bounded budget.**
>
> We agree to explicitly clarify the backbone's structural guarantees in the revision. These characterize the exponential backbone (not the learned dynamic subgraph):
>
> **Proposition 1 (Sparsity and diameter).** Each agent has ⌈log N⌉ candidates, yielding O(N logN) total edges and diameter ⌈log(N−1)⌉, ensuring O(log N)-hop reachability.
>
> **Proposition 2 (Spectral gap, adapted from [1]).** For the mixing matrix $W^{exp}$, 1−ρ($W^{exp}$) = 2/(1+⌈log N⌉) when N is even and < 2/(1+⌈log N⌉) when N is odd. Backbone connectivity thus degrades only logarithmically with N, justifying its use as a scalable communication prior.
>
> [1] Ying, B., et al., Exponential graph is provably efficient for decentralized deep training. Advances in Neural Information Processing Systems, 2021.
>
> **Q3: Applicability beyond CTDE**
>
> Yes. We believe the method can extend beyond the CTDE framework, as its core components are not inherently dependent on centralized training. **A plausible extension is that each agent could instead reuse neighbor embeddings from the previous communication round for local scoring, enabling fully decentralized training**. Extending SOPS to fully decentralized training paradigms is a natural future direction that we plan to explore.

---

> > ### Author Rebuttal · Reviewer_4mGF · 2026-04-03
> >
> > Thanks for the rebuttal. Regarding the Weaknesses and the three questions, the authors have provided responses, yet these explanations remain insufficient and ineffective.
> > I still have the following questions:
> > 1. I remain skeptical about large-scale training under the CTDE framework. Centralized training not only demands significant bandwidth but also relies on a powerful central server for computational support. It is unclear to me what the prospects are for researching scalability within this centralized framework, and I am curious about which specific scenarios can actually accommodate such large-scale centralized training.
> > 2. The authors claim that the proposed method can be integrated into a decentralized framework. Why was this not attempted? Including corresponding experimental results would make the paper significantly more convincing.

---

> > > ### Author Response · Authors · 2026-04-07
> > >
> > > We thank the reviewer for the continued evaluation and helpful comments. Our point-by-point responses follow.
> > >
> > > **Q1: Prospects for researching scalability within the CTDE framework**
> > >
> > > **a) Scalability challenges and opportunities within the CTDE framework:** CTDE's centralized training occurs entirely on the training server/simulator side — environments run parallel rollouts on GPUs, and the trainer updates shared network parameters from collected trajectories. **The "centralized" aspect refers to the ability to access global information (e.g., global state for the mixing network) during the learning phase**, not a requirement for real-time, high-bandwidth inter-agent communication.
> > >
> > > That said, **you correctly identify that large-scale CTDE training places significant pressure on the central server's GPU memory and computation**. This is precisely why researching scalable communication within the CTDE framework is valuable: **an efficient communication design directly reduces the central server's resource burden and extends the maximum trainable scale on the same hardware.** To substantiate this, we measured the GPU memory consumption of some methods on the IMP benchmark under N = 100, 200, 400, and 800 agents:
> > >
> > > ||Correlated-100|Correlated-200|Correlated-400|Correlated-800|
> > > |---|---|---|---|---|
> > > |SOPS|5370 MB|10468 MB|26302 MB|66742 MB|
> > > |EC-S|5124 MB|9670 MB|23242 MB|60938 MB|
> > > |EC-O|4988 MB|9355 MB|20140 MB|55832 MB|
> > > |GACG|10210 MB|22480 MB|54175 MB|OOM|
> > >
> > > Although SOPS incurs slightly higher GPU memory than the static-topology baselines, this modest overhead buys task-adaptive link selection that translates into consistently better returns. **We also added new experiments on IMP with N = 200 and N = 400 to further confirm that SOPS maintains its performance advantage at these larger scales** (metric: average return ± std, improvement over GACG):
> > >
> > > ||Correlated-200 ↓|Correlated-400 ↓|
> > > |---|---|---|
> > > |SOPS|-191.25±27.54 (+32.53%)|-259.44±31.97 (+30.04%)|
> > > |EC-S|-231.24±34.11 (+18.42%)|-294.57±42.28 (+20.57%)|
> > > |EC-O|-246.86±40.48 (+12.91%)|-318.65±35.04 (+14.07%)|
> > > |GACG|-283.42±57.30|-370.82±55.49|
> > >
> > > **b) Advantages of CTDE for large-scale MARL:** Compared with fully decentralized training, **CTDE offers practical advantages: access to global state enables more effective credit assignment and value decomposition, significantly improving sample efficiency and training stability**. Typical examples include **power-grid operation, large-scale navigation/robotics, and industrial process control**. The table below lists additional representative large-scale benchmarks and environments commonly used under CTDE beyond those included in our study:
> > >
> > > | Work | Domain | #Agents |
> > > | ------------- | ------------ | ------- |
> > > | EnEnv 1.0 [1] | Power system | 37      |
> > > | Grid2op [2]   | Power system | 100s    |
> > > | POGEMA [3]    | Navigation   | 256     |
> > > **In summary, by addressing CTDE's communication scalability bottleneck, our work lets practitioners retain its advantages in credit assignment and sample efficiency while scaling to large agent populations with strong performance.**
> > >
> > > [1] Bogucki, D. J., et al., 2025. EnEnv 1.0: Energy Grid Environment for Multi-Agent Reinforcement Learning Benchmarking. Proceedings of the 24th AAMAS.
> > >
> > > [2] B. Donnot, 2020. Grid2op - A testbed platform to model sequential decision making in power systems.
> > >
> > > [3] Skrynnik, A., et al., 2025. POGEMA: A Benchmark Platform for Cooperative Multi-Agent Pathfinding. ICLR.
> > >
> > > **Q2: Decentralized Implementation**
> > >
> > > Thank you for this suggestion. To address this concern of decentralized training more directly, **we additionally implemented a decentralized training and execution (DTE) proof-of-concept on top of IQL (a classical independent-learning baseline for DTE) and evaluated it on AdversarialPursuit with 45 and 61 agents** (metric: average return). In this variant, agents share network parameters due to homogeneity, but communication decisions are made locally: each agent maintains a small cache of the most recently received embeddings for its exponential-backbone candidates, scores candidate links using its own hidden state together with these cached neighbor embeddings and edge-type information, and selects a sparse set of communication partners at each step. No mixer, centralized critic, or centrally computed pairwise scorer is used.
> > >
> > > ||AdversarialPursuit-45|AdversarialPursuit-61|
> > > |---|---|---|
> > > |SOPS|71.68±7.12|78.74±6.47|
> > > |EC-O|61.23±8.05|66.91±7.44|
> > > |EC-S|42.56±8.67|47.18±8.21|
> > > |IQL|17.92±11.84|21.36±9.70|
> > >
> > > **The added results show that SOPS remains the best-performing method in this decentralized setting, outperforming EC-O, EC-S, and IQL on both scenarios.** It provides direct evidence that the benefit of topology-constrained adaptive sparse communication is not limited to CTDE. We will revise the paper to clarify this scope and present decentralized training as an additional validation rather than a claim of the current main method.

---

### Decision · Program_Chairs · 2026-04-30

**Decision:**

Accept (regular)

**Comment:**

The work addresses communication and scalability bottleneck in Multi-Agent Reinforcement Learning by proposing a scalable multi-agent communication framework utilizing exponential-graph backbone. I enjoyed reading this work. All reviewers found the work validating the claims well with the experimental results. Some of the key concerns regarding attention baselines and message dropout are addressed well by authors in the rebuttal. Following the reviewers consensus, I recommend a weak accept.